# Best-of-Both-Worlds Policy Optimization for CMDPs with Bandit Feedback

## Abstract

We study online learning in *constrained Markov decision processes* (CMDPs) in which rewards and constraints may be either *stochastic* or *adversarial*. In such settings, Stradi et al. (2024b) proposed the first *best-of-both-worlds* algorithm able to seamlessly handle stochastic and adversarial constraints, achieving optimal regret and constraint violation bounds in both cases. This algorithm suffers from two major drawbacks. First, it only works under *full feedback*, which severely limits its applicability in practice. Moreover, it relies on optimizing over the space of occupancy measures, which requires solving convex optimization problems, an highly inefficient task. In this paper, we provide the first *best-of-both-worlds* algorithm for CMDPs with *bandit feedback*. Specifically, when the constraints are *stochastic*, the algorithm achieves $\widetilde{\mathcal{O}}(\sqrt{T})$ regret and constraint violation, while, when they are *adversarial*, it attains $\widetilde{\mathcal{O}}(\sqrt{T})$ constraint violation and a tight fraction of the optimal reward. Moreover, our algorithm is based on a policy optimization approach, which is much more efficient than occupancy-measure-based methods.

## 1 Introduction

Most of the learning problems arising in real-world scenarios involve an agent sequentially interacting with an unknown environment. *Markov decision processes* (MDPs) (Puterman, 2014) have emerged as the most natural models for such interactions, as they allow to capture the fundamental goal of learning an optimal (*i.e.*, reward-maximizing) action-selection policy for the agent. However, in most of the real-world applications of interest, the learner also has to satisfy some additional requirements. For instance, in autonomous driving one has to avoid crashing with other cars (Isele et al., 2018), in ad auctions one must *not* deplete its allocated budget (He et al., 2021), while in recommendation systems offending items should *not* be presented to the users (Singh et al., 2020). In order to capture such requirements, *constrained MDPs* (CMDPs) (Altman, 1999) have been introduced. These augment classical MDPs by adding costs that the agent is constrained to keep below some given thresholds.

Over the last years, *online learning* problems in *episodic* CMDPs have received a growing attention (see, *e.g.*, (Efroni et al., 2020) for a seminal work in the field). These are problems in which the learner repeatedly interacts with the CMDP environment over multiple episodes. In such settings, the learner's goal is to minimize the *regret* of not always selecting a best-in-hindsight policy that satisfies cost constraints, while at the same time ensuring that the cumulative *violation* of cost constraints does *not* grow too fast over the episodes. Ideally, one would like that both the regret and the constraint violation grow sublinearly in the number of episodes $T$.

In online learning in episodic MDPs, two different assumptions on how rewards and costs are determined at each episode are possible. They can be selected either *stochastically* according to fixed (unknown) probability distributions or *adversarially*, meaning that no statistical assumption is made. Very recently, Stradi et al. (2024b) proposed the first *best-of-both-worlds* learning algorithm for online learning in episodic CMDPs. Such an algorithm is able to seamlessly handle stochastic and adversarial constraints, achieving optimal regret and violation bounds in both cases. However, it suffers from two major drawbacks. First, it only works under *full feedback*, meaning that the learning agent needs to observe rewards and costs defined over the whole environment after each episode. This is extremely unreasonable in practice, where only some feedback along the realized trajectory is usually available. Moreover, the algorithm works by optimizing over the space of occupancy measures, which requires solving a convex problem at every episode, an highly inefficient task.

## 1.1 ORIGINAL CONTRIBUTIONS

We provide the first *best-of-both-worlds* algorithm for online learning in episodic CMDPs with *bandit feedback*. This means that, after each episode, the algorithm only needs to observe the realized rewards and costs along the trajectory traversed during that episode, as it is the case in most of the real-world applications. Moreover, our algorithm is based on a primal-dual policy optimization method, and, thus, it is arguably much more efficient than the algorithm by Stradi et al. (2024b).

When the costs are *stochastic*, our algorithm achieves $\widetilde{O}(\sqrt{T})$ regret and constraint violation, while, when the costs are *adversarial*, it attains $\widetilde{O}(\sqrt{T})$ violation and a fraction of the optimal reward. These results match those of the full-feedback algorithm by Stradi et al. (2024b) and are provably tight.

We also analyze the performances of our algorithm with respect to a parameter $\rho$ measuring by "how much" Slater's condition is satisfied. Specifically, if $\rho$ is arbitrarily small, our algorithm can still guarantee $\widetilde{\mathcal{O}}(T^{3/4})$ regret and violation in the stochastic setting. Crucially, similarly to the algorithm by Stradi et al. (2024b), ours *does not require* any knowledge of the Slater's parameter $\rho$. In order to attain the aforementioned result, we show that the Lagrangian multipliers are automatically bounded during the learning dynamics, by employing the *no-interval-regret* property of our primal and dual regret minimizers. Indeed, we develop the first algorithm for unconstrained MDPs with no-interval-regret, under *bandit* feedback. This result may be of independent interest.

Finally, differently from Stradi et al. (2024b), our algorithm can achieve $\widetilde{\mathcal{O}}(\sqrt{T})$ regret and violation in the adversarial setting, by using a *weaker* baseline that has to satisfy the constraints at every round.

## 1.2 RELATED WORKS

In the following, we highlight the works that are mainly related to ours. Due to space constraints, we refer to Appendix A for a complete discussion about related works.

Online learning in MDPs has been widely studied both under stochastic settings (see (Auer et al., 2008)) and adversarial ones (see (Neu et al., 2010)). In adversarial settings, two feedbacks are usually investigated. In the *full-feedback setting*, the reward function (or loss) is entirely revealed at the end of the episode. In this case, Rosenberg & Mansour (2019b) show that it is possible to achieve an optimal $\widetilde{\mathcal{O}}(\sqrt{T})$ regret bound. In the more challenging *bandit-feedback setting*, with rewards revealed along the traversed trajectory only, Jin et al. (2020) show that the optimal bound is still attainable.

As concerns MDPs with constraints, online learning has been studied mainly in the stochastic setting (see Efroni et al. (2020) for a seminal work on the topic). As concerns adversarial settings, namely, when the constraints are *not* assumed to be stochastic, there exists an impossibility result from Mannor et al. (2009) that prevents from attaining sublinear regret and violation when the optimal solution is computed with respect to a policy that satisfies the constraints on average. Thus, many works focused on achieving $\widetilde{\mathcal{O}}(\sqrt{T})$ regret and violation for adversarial rewards and stochastic constraints (see (Qiu et al., 2020)) or non-stationary environments with bounded non-stationarity (see (Ding & Lavaei, 2023; Wei et al., 2023; Stradi et al., 2024c)).

Recently, Stradi et al. (2024b) showed the first best-of-both-worlds (with respect to the constraints) algorithm for CMDPs. Precisely, the authors propose a primal-dual algorithm that optimizes over the occupancy measure space, under *full feedback*. When the constraints are stochastic, the algorithm achieves $\widetilde{\mathcal{O}}(\sqrt{T})$ regret and violation, both in the case in which rewards are adversarial and the one where they are stochastic. Contrariwise, in the adversarial setting, the algorithms attains $\widetilde{\mathcal{O}}(\sqrt{T})$ violatios, and the no-$\alpha$-regret property with $\alpha = \rho/H+\rho$, where $\rho$ is a suitably-defined Slater's parameter. Notice that this result is in line with the best-of-both-worlds results in the single-state online constrained settings, *e.g.*, (Castiglioni et al., 2022b).

## 2 PROBLEM SETTING

In this section, we describe the problem setting and the related notation.

## 2.1 ONLINE CONSTRAINED MARKOV DECISION PROCESSES

We study *online episodic constrained* MDPs (Altman, 1999) (CMDPs), which are defined as tuples $M := (X, A, P, \{r_t\}_{t=1}^T, \{G_t\}_{t=1}^T)$. Specifically, $T$ is a number of episodes, with $t \in [T]$ denoting a specific episode. $X, A$ are finite state and action spaces, respectively. $P : X \times A \to \Delta(X)$ is the transition function. We denote by $P(x'|x, a)$ the probability of going from state $x \in X$ to $x' \in X$ by taking action $a \in A$. Notice that, w.l.o.g., in this work we consider *loop-*

---

**Algorithm 1** Learner-Environment Interaction

1: **for** $t = 1, \ldots, T$ **do**
2:      $r_t$ and $G_t$ are chosen *stochastically* or *adversarially*
3:      The learner chooses a policy $\pi_t : X \times A \to [0, 1]$
4:      The state is initialized to $x_0$
5:      **for** $h = 0, \ldots, H - 1$ **do**
6:          The learner plays $a_h \sim \pi_t(\cdot|x_h)$
7:          The learner observes $r_t(x_h, a_h)$ and $g_{t,i}(x_h, a_h)$ for all $i \in [m]$
8:          The environment evolves to $x_{h+1} \sim P(\cdot|x_h, a_h)$
9:          The learner observes $x_{h+1}$

---

*free* CMDPs. Formally, this means that $X$ is partitioned into $H$ layers $X_0, \ldots, X_H$ such that the first and the last layers are singletons, *i.e.*, $X_0 = \{x_0\}$ and $X_H = \{x_H\}$, and that $P(x'|x, a) > 0$ only if $x' \in X_{h+1}$ and $x \in X_h$ for some $h \in [0 .. H - 1]$. We observe that any episodic CMDP with horizon $H$ that is *not* loop-free can be cast into a loop-free one by suitably duplicating the state space $H$ times, *i.e.*, a state $x$ is mapped to a set of new states $(x, h)$, where $h \in [0 .. H]$. $\{r_t\}_{t=1}^T$ is a sequence of vectors describing the rewards at each episode $t \in [T]$, namely $r_t \in [0, 1]^{|X \times A|}$. We refer to the reward of a specific state-action pair $x \in X, a \in A$ for an episode $t \in [T]$ as $r_t(x, a)$. Rewards may be either *stochastic*, in that case $r_t$ is a random variable distributed according to a distribution $\mathcal{R}$ for every $t \in [T]$, or chosen by an *adversary*. $\{G_t\}_{t=1}^T$ is a sequence of constraint matrices describing the $m$ *constraint* violations at each episode $t \in [T]$, namely $G_t \in [-1, 1]^{|X \times A| \times m}$, where non-positive violation values stand for satisfaction of the constraints. For $i \in [m]$, we refer to the violation of the $i$-th constraint for a specific state-action pair $x \in X, a \in A$ at episode $t \in [T]$ as $g_{t,i}(x, a)$. Constraint violations may be *stochastic*, in that case $G_t$ is a random variable distributed according to a probability distribution $\mathcal{G}$ for every $t \in [T]$, or chosen by an *adversary*.

In the online setting, the learner chooses a *policy* $\pi : X \to \Delta(A)$ at each episode, defining a probability distribution over actions at each state. For ease of notation, we denote by $\pi(\cdot|x)$ the probability distribution for a state $x \in X$, with $\pi(a|x)$ denoting the probability of action $a \in A$. In Algorithm 1 we provide the interaction between the learner and the environment in a CMDP. Furthermore, we assume that the learner knows $X$ and $A$, but they do *not* know anything about $P$. Notice that the interaction between the learner and the environment is with *bandit feedback*, namely, the rewards and the constraint violations are revealed for the traversed trajectory only.

**Occupancy Measures** Given a transition function $P$ and a policy $\pi$, the *occupancy measure* $q^{P,\pi} \in [0, 1]^{|X \times A \times X|}$ induced by $P$ and $\pi$ is such that, for all $x \in X_h$, $a \in A$, and $x' \in X_{h+1}$ with $h \in [0 .. H - 1]$, it holds $q^{P,\pi}(x, a, x') = \mathbb{P}\{x_h = x, a_h = a, x_{h+1} = x'|P, \pi\}$. Moreover, we also define $q^{P,\pi}(x, a) = \sum_{x' \in X_{h+1}} q^{P,\pi}(x, a, x')$ and $q^{P,\pi}(x) = \sum_{a \in A} q^{P,\pi}(x, a)$. Then, following (Rosenberg & Mansour, 2019b), the set of *valid* occupancy measures can be characterized as follows. Specifically, $q \in [0, 1]^{|X \times A \times X|}$ is a valid occupancy measure of an episodic loop-free MDP if and only if the following conditions hold: (i) $\sum_{x \in X_h} \sum_{a \in A} \sum_{x' \in X_{h+1}} q(x, a, x') = 1$ for all $h \in [0, \ldots, H - 1]$; (ii) $\sum_{a \in A} \sum_{x' \in X_{h+1}} q(x, a, x') = \sum_{x' \in X_{h-1}} \sum_{a \in A} q(x', a, x)$ for all $h \in [1, \ldots, H - 1]$ and $x \in X_h$; and (iii) $P^q = P$, where $P$ is the transition function of the MDP and $P^q$ is the one induced by $q$. Indeed, any valid occupancy measure $q$ induces a transition function $P^q$ and a policy $\pi^q$, defined as $P^q(x'|x, a) := \frac{q(x, a, x')}{q(x, a)}$ and $\pi^q(a|x) := \frac{q(x, a)}{q(x)}$.

## 2.2 OFFLINE CMDPs BASELINE

In the following, we introduce the *offline* CMDP optimization problem, which is needed to define a proper baseline to evaluate the performances of online learning algorithms. Specifically, we introduce the following linear program parameterized by a reward vector $r$ and a constraint matrix $G$ as follows:

$$\text{OPT}_{r,G} := \begin{cases} \max_{q \in \Delta(M)} & r^\top q \\ \text{s.t.} & G^\top q \le \underline{0}, \end{cases} \tag{1}$$

where $q \in [0, 1]^{|X \times A|}$ is an occupancy measure and $\Delta(M)$ is the set of valid occupancy measures.

Furthermore, we state the following well-known condition on the offline CMDP problem.

**Condition 1** (Slater's condition). *Given a constraint matrix $G$, the Slater's condition holds when there is a strictly feasible solution $q^\diamond$ such that $G^\top q^\diamond < \underline{0}$.*

Notice that, in this work, we do *not* assume that the Slater's condition holds. Indeed, our algorithm still works when a strictly feasible solution does *not* exists. We refer to Section 2.4 for further details on this. Finally, we define the Lagrangian function of Problem (1), as follows.

**Definition 1** (Lagrangian function). *Given a reward vector $r$ and a constraint matrix $G$, the Lagrangian function $\mathcal{L}_{r,G} : \Delta(M) \times \mathbb{R}_{\geq 0}^m \to \mathbb{R}$ of Problem (1) is defined as:*

$$\mathcal{L}_{r,G}(q, \lambda) := r^\top q - \lambda^\top(G^\top q).$$

### 2.3 ONLINE LEARNING PROBLEM

As it is standard in the online learning literature (Cesa-Bianchi & Lugosi, 2006), we evaluate the performance of learning algorithms by means of the notion of cumulative regret.

**Definition 2.** *We define the cumulative regret up to episode $T$ as:*

$$R_T := T\,\mathrm{OPT}_{\overline{r},\overline{G}} - \sum_{t=1}^{T} r_t^\top q^{P,\pi_t},$$

*where $\overline{r} := \mathbb{E}_{r\sim\mathcal{R}}[r]$ if rewards are stochastic and $\overline{r} := \frac{1}{T}\sum_{t=1}^{T} r_t$ if they are adversarial, while $\overline{G} := \mathbb{E}_{G\sim\mathcal{G}}[G]$ if the constraints are stochastic and $\overline{G} := \frac{1}{T}\sum_{t=1}^{T} G_t$ if they are adversarial.*

We refer to an optimal occupancy measure, *i.e.*, a feasible one achieving value $\mathrm{OPT}_{\overline{r},\overline{G}}$, as $q^*$. Thus, we can rewrite the regret definition as $R_T = \sum_{t=1}^{T} \overline{r}^\top q^* - \sum_{t=1}^{T} r_t^\top q^{P,\pi_t}$. Notice that, in the adversarial setting, the regret is computed with respect to an *optimal feasible strategy in hindsight*. Indeed, an optimal solution is *not* required to satisfy the constraints at every episode $t \in [T]$.

Next, we define the performance measure related to constraints: the *cumulative constraint violation*.

**Definition 3.** *The cumulative constraint violation up to episode $T$ is defined as:*

$$V_T := \max_{i\in[m]} \sum_{t=1}^{T} \left[ G_t^\top q^{P,\pi_t} \right]_i.$$

Learning algorithms perform properly when they are capable of keeping both the quantities defined above sublinear in $T$, namely, $R_T = o(T)$ and $V_T = o(T)$.

For the sake of simplicity, in the rest of the paper, we will refer to $q^{P,\pi_t}$ as $q_t$, omitting the dependence on transition unction $P$ and policy $\pi$.

### 2.4 FEASIBILITY

We introduce a problem-specific parameter of Problem (1), called $\rho \in [0, H]$, which identifies by "how much" Slater's condition is satisfied. Formally:

- when the constraints are selected *stochastically*, namely, they are chosen from a fixed distribution, the parameter $\rho$ is defined as $\rho := \max_{q\in\Delta(M)} \min_{i\in[m]} - \left[ \overline{G}^\top q \right]_i$.

- when the constraints are chosen *adversarially*, namely, no statistical assumption is made, the parameter $\rho$ is defined as $\rho := \max_{q\in\Delta(M)} \min_{t\in[T]} \min_{i\in[m]} - \left[ G_t^\top q \right]_i$.

Furthermore, we denote the occupancy measure $q \in \Delta(M)$ leading to the value of $\rho$ by $q^\circ$. Intuitively, $\rho$ represents the "margin" by which the "most feasible" strictly feasible solution (*i.e.*, $q^\circ$) satisfies the constraints. Finally, we state the following condition on the parameter $\rho$, which will guide the analyses of the performances of our algorithm.

**Condition 2.** *It holds that $\rho \geq T^{-\frac{1}{8}} H \sqrt{112m}$.*

Notice that, it is standard in the literature (see, *e.g*, (Efroni et al., 2020)) to assume that $\rho$ is a constant independent of $T$ and directly include in the regret bound the dependence on $1/\rho$. Nevertheless, when $\rho$ is too small, this could result in suboptimal regret bounds. In this paper, we take a different approach by providing theoretical guarantees for any value of $\rho$.

## 3 A Policy Optimization Primal-Dual Approach

In this section, we provide the description of our algorithm. We resort to a primal-dual formulation of the CMDP problem, and we employ different regret minimizers to optimize over the primal space (namely, the policy space) and the dual one (that is, the Lagrangian variables space). Furthermore, our primal algorithm is based on a policy optimization approach. Thus, the learning update is *not* performed over the occupancy measure space, but state-by-state along the MDP structure. This allows us to avoid solving a convex program at each episode (as it is the case in the algorithm by (Stradi et al., 2024b)). As concerns the dual, we employ online gradient descent (OGD). We remark that our algorithm does *not* require any knowledge of the Slater's parameter $\rho$. Indeed, as we further discuss in the rest of this work, we can show that the Lagrangian multipliers are automatically bounded given specific no-regret properties of the primal and dual regret minimizers.

### 3.1 Meta-Algorithm

In Algorithm 2, we provide the pseudocode of *primal-dual bandit policy search* (PDB-PS).

---
**Algorithm 2** PDB-PS

**Require:** State space $X$, action space $A$, number of episodes $T$, confidence parameter $\delta \in (0, 1)$
1: $\pi_1(a|x) \leftarrow \frac{1}{|A|} \quad \forall (x, a) \in X \times A$
2: $\lambda_1 \leftarrow \underline{0}, \Gamma_1 \leftarrow 1, \Xi_1 \leftarrow 2, \mathcal{K} \leftarrow \left[0, T^{1/4}\right]^m, \eta \leftarrow \frac{1}{D \ln\left(|A||X|^2 T^2/\delta\right)\sqrt{T}}$
3: **for** $t = 1, \dots, T$ **do**
4:     Play policy $\pi_t$, observe trajectory $\{(x_h, a_h)\}_{h=0}^{H-1}$, rewards $\{r_t(x_h, a_h)\}_{h=0}^{H-1}$ and violations $\{g_{t,i}(x_h, a_h)\}_{h=0}^{H-1}$ for all $i \in [m]$
5:     **for** $h = 0, \dots, H - 1$ **do**
6:         $\ell_t(x_h, a_h) \leftarrow \Gamma_t + \sum_{i=1}^m \lambda_{t,i} g_{t,i}(x_h, a_h) - r_t(x_h, a_h)$
7:     $\pi_{t+1} \leftarrow \text{FS-PODB.UPDATE}(\{(x_h, a_h)\}_{h=0}^{H-1}, \{\ell_t(x_h, a_h)\}_{h=0}^{H-1}, \Xi_t)$
8:     $\lambda_{t+1} \leftarrow \Pi_{\mathcal{K}}\left[\lambda_t + \eta \sum_{h=0}^{H-1} G_t[x_h, a_h]\right]$
9:     $\Gamma_{t+1} \leftarrow 1 + \|\lambda_{t+1}\|_1$
10:    $\Xi_{t+1} \leftarrow \max\{\Xi_t, 2\Gamma_t\}$

---

Algorithm 2 initializes the policy uniformly over the space (see Line 1). Moreover, the Lagrangian variables are initialized as the zero vector, the loss scaling factor to 1, the loss range to 2, and, finally, the dual space is instantiated as $\left[0, T^{1/4}\right]^m$ (see Line 2). We underline that we force the dual space to be bounded in $\left[0, T^{1/4}\right]^m$ only to deal with degenerate cases where Condition 2 does *not* hold. When Condition 2 holds, our algorithm guarantees that the Lagrangian variables are automatically bounded during learning. Furthermore, the algorithm keeps track of the maximum loss range observed by the primal algorithm $\Xi_t$, up to episode $t \in [T]$, since the primal regret minimizer needs to dynamically update its belief on the loss range, in order to attain optimal regret bounds. The algorithm plays policy $\pi_t$ and observes the *bandit* feedback as depicted in Algorithm 1 (see Line 4). Given the observed feedback, PDB-PS builds a re-scaled Lagrangian loss for each layer $h \in [H]$ as:

$$\ell_t(x_h, a_h) := \Gamma_t + \sum_{i=1}^m \lambda_{t,i} g_{t,i}(x_j, a_j) - r_t(x_j, a_j). \tag{2}$$

Notice that the loss built in Equation (2) can been seen as the Lagrangian suffered by $\pi_t$ for state-action pair $(x, a)$, scaled by $\Gamma_t$ to guarantee that the losses are always positive (see Line 6). This loss is properly built to feed the primal policy optimization procedure. Moreover, we underline that the feedback given to the primal algorithm encompasses the trajectory and the maximum loss range

observed, besides the loss built in Equation (2). Policy $\pi_{t+1}$ is returned by the primal algorithm (Line 7). We refer the reader to the next section for further discussion on the primal optimization algorithm. Algorithm 2 updates the Lagrangian multipliers using an online gradient descent update with loss $-\sum_{h=0}^{H} G_t[x_h, a_h]$ in the bounded dual space $[0, T^{1/4}]^m$ as:

$$\lambda_{t+1} \leftarrow \Pi_{\mathcal{K}} \left[ \lambda_t + \eta \sum_{h=0}^{H-1} G_t[x_h, a_h] \right],$$

where $\Pi_{\mathcal{K}}$ is the euclidean projection over the space $\mathcal{K}$ and $G_t[x_h, a_h]$ is the $m$-dimensional vector composed by the violations of any constraint for the state-action pair $(x_h, a_h)$ (Line 8). Thus, the current loss scaling factor is computed as $\Gamma_{t+1} \leftarrow 1 + \|\lambda_{t+1}\|_1$ (Line 9). Finally, the maximum observed loss range $\Xi_{t+1}$ is updated as $\Xi_{t+1} \leftarrow \max\{\Xi_t, 2\Gamma_{t+1}\}$, since the range of losses observed by the primal depends on the Lagrangian multipliers values (Line 10).

## 3.2 Primal Regret Minimizer

In Algorithm 3, we provide the pseudocode of *fixed share policy optimization with dilated bonus* (`FS-PODB.UPDATE`), namely, the update performed by the primal regret minimizer employed by Algorithm 2. Algorithm 3 builds on top of the state-of-the-art policy optimization algorithm for adversarial MDPs (see (Luo et al., 2021)), equipping it with a fixed share update (Cesa-Bianchi et al., 2012). This modification allows us to achieve the no-interval regret property, which, to the best of our knowledge, has never been shown for adversarial MDPs with bandit feedback. Thus, we believe that the theoretical guarantees of Algorithm 3 are of independent interest.

Specifically, Algorithm 3 requires in input the trajectory traversed during the learner-environment interaction, the incurred loss functions, and the maximum loss range observed for any $t \in [T]$.[1] During the first episode, the algorithm initializes the estimated transitions space as the set of all possible transition functions (Line 2). Thus, at each episode the algorithm defines a dynamic learning rate $\eta_t \propto \frac{1}{\sqrt{T}\Xi_t}$ (Line 3), where $\Xi_t$ is the upper bound on the range of the loss functions up to $t$. This is done to control the different scales of the loss, due to the Lagrangian multipliers choice of the dual algorithm. Then, Algorithm 3 builds an *optimistic* estimator of the state-action value function as:

$$\widehat{Q}_t(x, a) \coloneqq \frac{L_{t,h}}{\overline{q}_t(x, a) + \gamma} \mathbb{I}_t(x, a),$$

where $\mathbb{I}_t(x, a) \coloneqq \mathbb{I}\{x_{t,h} = x, a_{t,h} = a\}$ and $L_{t,h} \coloneqq \sum_{j=h}^{H-1} \ell_t(x_j, a_j)$ is the loss incurred by the algorithm at episode $t$ starting from layer $h$. Indeed, since $\overline{q}_t(x, a) \coloneqq \max_{\widehat{P} \in \mathcal{P}_t} q^{\widehat{P}, \pi_t}(x, a)$,[2] and $\gamma$ is a positive quantity, $\widehat{Q}_t(x, a)$ results in an optimistic estimator of the state-action value function (Line 4). The optimistic estimator is employed to control the variance of the loss estimation and, thus, in order to achieve high-probability results. Finally, notice that the state-action value function (as the estimated one) is commonly used in policy optimization as it allows to optimize efficiently state-by-state. In addition to the estimated state-action value function, Algorithm 3 defines a dilated bonus similar to the one introduced by Luo et al. (2021), which is then incorporated in the final objective of the optimization update. The bonus is defined as:

$$B_t(x, a) \coloneqq b_t(x) + \left(1 + \frac{1}{H}\right) \max_{\widehat{P} \in \mathcal{P}_t} \mathbb{E}_{x' \sim \widehat{P}(\cdot|x,a)} \mathbb{E}_{a \sim \pi_t(\cdot|x')} \left[B_t(x', a')\right],$$

where the term $b_t(x)$ depends on the uncertainty on the transitions estimation and the range of the losses, while the term $\left(1 + \frac{1}{H}\right)$ attributes more weight to the deeper layers, so as to incentivize exploration (Line 5). The weights associated to any action are computed employing the so called fixed share update (Cesa-Bianchi et al., 2012); specifically, the weights are computed as the convex combination between the uniform weight and the solution to optimization step $\propto w_t(a|x) e^{-\eta_t \left(\widehat{Q}_t(x,a) - B_t(x,a)\right)}$. The policy is simply computed as a normalization between weights (see Line 6). Notice that the

---

[1] Notice that, while the input of Algorithm 3 may seem different from the standard bandit feedback received in adversarial MDPs, this is *not* the case. Indeed, it is sufficient to set $\Xi_t = 1$ for all $t \in [T]$ to achieve the same guarantees attained by Algorithm 3, in the Lagrangian formulation of CMDPs, in standard adversarial MDPs.

[2] As shown in (Jin et al., 2020), $\overline{q}_t(x, a)$ can be computed efficiently by means of dynamic programming.

---

**Algorithm 3** FS-PODB.UPDATE

---

**Require:** Trajectory $\{(x_h, a_h)\}_{h=0}^{H-1}$, losses $\{\ell_t(x_h, a_h)\}_{h=0}^{H-1}$, loss range upper bound $\Xi_t$

1: **if** $t = 1$ **then**
2: $\quad \mathcal{P}_1 \leftarrow$ set of all possible transitions
3: $\eta_t \leftarrow \frac{1}{2H\Xi_t C\sqrt{T}}, \gamma \leftarrow \frac{1}{C\sqrt{T}}, \sigma \leftarrow \frac{1}{T}$
4: For all $h = 0, \ldots, H - 1$ and $(x, a) \in X_h \times A$:

$$L_{t,h} \leftarrow \sum_{j=h}^{H-1} \ell_t(x_j, a_j)$$

$$\widehat{Q}_t(x, a) \leftarrow \frac{L_{t,h}}{\overline{q}_t(x, a) + \gamma} \mathbb{I}_t(x, a),$$

$\quad$ where $\overline{q}_t(x, a) := \max_{\widehat{P} \in \mathcal{P}_t} q^{\widehat{P}, \pi_t}(x, a)$ and $\mathbb{I}_t(x, a) := \mathbb{I}\{x_{t,h} = x, a_{t,h} = a\}$
5: For all $(x, a) \in X \times A$:

$$b_t(x) \leftarrow \mathbb{E}_{a \sim \pi_t(\cdot|x)} \left[ \frac{3\gamma H\Xi_t + H\Xi_t \left( \overline{q}_t(x, a) - \underline{q}_t(x, a) \right)}{\overline{q}_t(x, a) + \gamma} \right]$$

$$B_t(x, a) \leftarrow b_t(x) + \left( 1 + \frac{1}{H} \right) \max_{\widehat{P} \in \mathcal{P}_t} \mathbb{E}_{x' \sim \widehat{P}(\cdot|x,a)} \mathbb{E}_{a \sim \pi_t(\cdot|x')} [B_t(x', a')]$$

$\quad$ where $\underline{q}_t(x, a) := \min_{\widehat{P} \in \mathcal{P}_t} q^{\widehat{P}, \pi_t}(x, a)$, and $B_t(x_H, a) := 0$ for all $a \in A$
6: For all $(x, a) \in X \times A$:

$$w_{t+1}(a|x) \leftarrow (1 - \sigma) w_t(a|x) e^{-\eta_t \left( \widehat{Q}_t(x,a) - B_t(x,a) \right)} + \frac{\sigma}{|A|} \sum_{a' \in A} w_t(a'|x) e^{-\eta_t \left( \widehat{Q}_t(x,a') - B_t(x,a') \right)}$$

$$\pi_{t+1}(a|x) \leftarrow \frac{w_{t+1}(a|x)}{\sum_{a' \in A} w_{t+1}(a'|x)}$$

7: $\mathcal{P}_{t+1} \leftarrow$ TRANSITION.UPDATE$(\{(x_h, a_h)\}_{h=0}^{H-1})$

---

convex combination mentioned above is crucial to bound the regret for each interval (that is, to attain the no-interval regret property). Indeed, it guarantees a lower bound for the value taken by the policy in each available action at each episode, and, thus, for all intervals $[t_1, t_2] \subset [T]$, it allows to find a nice upper bound for the Bregman divergence $D_\psi(\pi(\cdot|a); \pi_{t_1}(\cdot|a))$, for all policies $\pi$. Finally, the estimation of the transitions is updated given the trajectory traversed in the MDP (Line 7). This estimation is standard in the literature. Thus, we refer to (Rosenberg & Mansour, 2019b) for further discussion on the use of counters and epochs to estimate a superset of the transition space $\mathcal{P}_t$.

### 3.3 NO-INTERVAL REGRET PROPERTY

When the Slater's parameter $\rho$ is known, the only necessary requirement for the primal and the dual regret minimizers is to be no-regret. Thus, it is sufficient to bound the Lagrangian space so that $\|\lambda\|_1 \leq \mathcal{O}(H/\rho)$ to attain sublinear regret and violation. Nevertheless, knowing $\rho$ is generally *not* possible in real-world scenarios. In order to relax the assumption on the knowledge of $\rho$, we require our primal and dual regret minimizers to have the no-interval regret property.[3]

First, we introduce the interval regret as follows.

**Definition 4** (Interval regret). *Given an interval of consecutive episodes* $[t_1, \ldots, t_2] \subseteq [1, \ldots, T]$, *the* interval regret *with respect to a general occupancy q (and the associated policy $\pi$) and a sequence of loss functions* $\{\ell_t\}_{t=1}^T$ *with* $\ell_t : X \times A \to [0, K]$, *with* $K > 0$, *is* $R_{t_1, t_2}(q) := \sum_{t=t_1}^{t_2} \ell_t^\top (q_t - q)$.

---

[3]What we require is generally known in the literature as the *weak* no-interval regret property. For the sake of simplicity, in our work, we introduce only the *weak* property.

In the following, we omit the dependence on the general occupancy $q$ when it is clear from the context. Thus, given Definition 4, we are able to introduce the no-interval regret property.

**Definition 5** (No-interval regret property). *An algorithm attains the no-interval regret property when for any interval of consecutive episodes $[t_1, \ldots, t_2] \subseteq [1, \ldots, T]$ and with respect to any valid occupancy $q$ (and the associated policy $\pi$), it holds $R_{t_1, t_2} \leq \widetilde{\mathcal{O}}(\sqrt{T})$.*

Intuitively, the no-interval regret property guarantees a more stable learning dynamics over the episodes. When *full feedback* is available, as for the dual algorithm, it is sufficient to employ `OGD`-like updates to attain the desired result. This is *not* the case when the feedback is *bandit*. Nevertheless, given that we use a policy optimization procedure and the fixed share update, we build the first algorithm for adversarial MDPs with no-interval-regret. We state the result in the following theorem.

**Theorem 3.** *For any $\delta \in (0, 1)$, with probability at least $1 - 8\delta$, Algorithm `FS-PODB` attains:*

$$R_{t_1, t_2} \leq \widetilde{\mathcal{O}}\left(\Xi_{t_1, t_2}\sqrt{T} + \Xi_{t_1, t_2}\frac{t_2 - t_1}{\sqrt{T}}\right),$$

*where the regret can be computed with respect to any policy function $\pi : X \to \Delta(A)$.*

Notice that, as it is standard for online learning algorithms, $R_{t_1, t_2}$ scales as the loss range, as shown by the dependence on $\Xi_{t_1, t_2}$, that is, the maximum possible range of losses in the interval.

### 3.4 Bound on the Lagrangian Multipliers Dynamics

Next, we show that, given the no-interval regret property of the primal and the dual regret minimizers, it is possible to show that the Lagrangian multipliers are automatically bounded during learning. Notice that this bound is necessary since any adversarial regret minimizer needs the loss to be bounded to achieve the no-regret property. Thus, since the rewards $\{r_t\}_{t=1}^T$ and the constraints $\{G_t\}_{t=1}^T$ are assumed to be bounded for all episodes, the problem of bounding the loss suffered by the primal algorithm becomes the problem of bounding the Lagrangian multipliers $\{\lambda_t\}_{t=1}^T$.

**Theorem 4.** *Under Condition 2, for any $\delta \in (0, 1)$, with probability at least $1 - 11\delta$, it holds:*

$$\|\lambda_t\|_1 \leq \Lambda \quad \forall t \in [T + 1],$$

*where $\Lambda = \frac{112mH^2}{\rho^2}$.*

The general idea behind the proof is to compare, for every interval $[t_1, t_2] \subset [T]$, the upper bound to $-\sum_{t=t_1}^{t_2} \ell_t^{\mathcal{L}, \top} q_t$ obtained through the regret of the dual algorithm with the lower bound to the same quantity obtained through the primal interval regret, where we define the non-scaled Lagrangian loss $\ell_t^{\mathcal{L}}$ as the vector composed by $\ell_t^{\mathcal{L}}(x, a) \coloneqq \sum_{i=1}^m \lambda_{t,i} g_{t,i}(x, a) - r_t(x, a)$ for all $(x, a) \in X \times A$ and for all $t \in [T]$. The resulting inequality leads, by contradiction, to the desired bound. In this sense, a fundamental requirement for the proof is that the regret guarantees for both the primal and the dual algorithm hold for all subsets of episodes.

## 4 Theoretical Analysis

In this section, we prove the best-of-both-world guarantees attained by Algorithm 2.

### 4.1 Stochastic Setting

We first study the performance of Algorithm 2 when the constraints are stochastic.

In such a setting, our algorithm can handle two scenarios. In both of them, employing a primal-dual analysis shows that both the regret and the violations are bounded with order $\widetilde{\mathcal{O}}(\sqrt{T})$ times the maximum value taken over all episodes of the Lagrangian multipliers, *i.e.* $\max_{t \in [T]} \|\lambda_t\|_1$. In the first scenario, Condition 2 holds and thus we can apply Theorem 4 to show that the Lagrangian multipliers are bounded. In such a case, $\max_{t \in [T]} \|\lambda_t\|_1$ can be easily bounded by $\Lambda$. When Conditions 2 does *not* hold, we need to resort to the bound of Lagrangian multipliers derived by the instantiation of `OGD` decision space, leading to $\widetilde{\mathcal{O}}(T^{3/4})$ regret and violations bounds.

Specifically, when Condition 2 holds, the Lagrangian multipliers are nicely bounded by $\Lambda$.

**Theorem 5.** *Suppose that Condition 2 holds and the constraints are generated stochastically. Then, for any $\delta \in (0, 1)$, Algorithm 2 attains:*

$$R_T \leq \widetilde{\mathcal{O}}\left(\Lambda\sqrt{T}\right), \quad V_T \leq \widetilde{\mathcal{O}}\left(\Lambda\sqrt{T}\right),$$

*with probability at least $1 - 14\delta$ when the rewards are stochastic and at least $1 - 13\delta$ when the rewards are adversarial.*

When Condition 2 does *not* hold, we can still use the bound forced by Algorithm 2 on the dual space. Therefore, the Lagrangian multipliers are bounded by $mT^{1/4}$, leading to the following result.

**Theorem 6.** *Suppose that Condition 2 does not hold and the constraints are generated stochastically. Then, for any $\delta \in (0, 1)$, Algorithm 2 attains:*

$$R_T \leq \widetilde{\mathcal{O}}\left(T^{3/4}\right), \quad V_T \leq \widetilde{\mathcal{O}}\left(T^{3/4}\right),$$

*with probability at least $1 - 11\delta$ when the rewards are stochastic and at least $1 - 10\delta$ when the rewards are adversarial.*

### 4.2 ADVERSARIAL SETTING

We then study the performance of Algorithm 2 when the constraints are adversarial. Notice that, in such a setting, there exists an impossibility result from (Mannor et al., 2009) that prevents any algorithm from attaining both sulinear regret and sublinear violations. Thus, best-of-both-worlds algorithms in constrained settings focus on attaining sublinear violations and a fraction of the optimal rewards (see *e.g.*, (Castiglioni et al., 2022b; Stradi et al., 2024b)).[4]

In such a setting, we can show the following result.

**Theorem 7.** *Suppose Condition 2 holds and the constraints are adversarial. Then, for any $\delta \in (0, 1)$, Algorithm 2 attains:*

$$\sum_{t=1}^{T} r_t^\top q_t \geq \Omega\left(\frac{\rho}{\rho + H} \cdot OPT_{\overline{r}, \overline{G}}\right), \quad V_T \leq \widetilde{\mathcal{O}}\left(\Lambda\sqrt{T}\right),$$

*with probability at least $1 - 14\delta$ when the rewards are stochastic and with probability at least $1 - 13\delta$ when the rewards are adversarial.*

#### 4.2.1 A WEAKER BASELINE

In this section, we show that the impossibility result by Mannor et al. (2009) can be circumvented by adopting a different baseline in the regret definition. Precisely, we compute the weaker baseline as the solution to the following linear program:

$$\text{OPT}^{\mathcal{W}} := \begin{cases} \max_{q \in \Delta(M)} & \overline{r}^\top q \\ \text{s.t.} & G_t^\top q \leq \underline{0} \quad \forall t \in [T]. \end{cases}$$

Notice that, in the previous sections, we allow the optimal policy $q^*$ to satisfy the constraints on average, *i.e.*, $\sum_{t=1}^{T} G_t^\top q^* \leq 0$. In such a case, the set of feasible policies is much smaller than the one associated with the weaker baseline, that is, when a feasible policy must satisfy the constraints *at each episode*. Given the new baseline, we can rewrite the regret as $R_T := T \, \text{OPT}^{\mathcal{W}} - \sum_{t=1}^{T} r_t^\top q_t$.

When the regret is computed w.r.t. the weaker baseline, we can recover the same theoretical results of the stochastic setting. Precisely, when Condition 2 holds we have the following result.

**Theorem 8.** *Suppose that Condition 2 holds and the constraints are generated adversarially. Then, for any $\delta \in (0, 1)$, Algorithm 2 attains:*

$$R_T \leq \widetilde{\mathcal{O}}\left(\Lambda\sqrt{T}\right), \quad V_T \leq \widetilde{\mathcal{O}}\left(\Lambda\sqrt{T}\right),$$

*with probability at least $1 - 13\delta$ when the rewards are stochastic and at least $1 - 12\delta$ when the rewards are adversarial.*

---

[4]Attaining the no-$\alpha$-regret property, that is, being no-regret w.r.t. a fraction of the optimum, achieving a competitive ratio, and guaranteeing a fraction of the optimal rewards are used as synonyms in the literature, since any of the aforementioned guarantees can be derived by the others.

We conclude the section by analyzing the scenario in which Condition 2 does *not* hold.

**Theorem 9.** *Suppose that Condition 2 does not hold and the constraints are generated adversarially. Then, for any $\delta \in (0, 1)$, Algorithm 2 attains:*

$$R_T \leq \widetilde{\mathcal{O}}\left(T^{3/4}\right), \quad V_T \leq \widetilde{\mathcal{O}}\left(T^{3/4}\right),$$

*with probability at least $1 - 12\delta$ when the rewards are stochastic and at least $1 - 11\delta$ when the rewards are adversarial.*

Intuitively, Theorems 8 and 9 can be proved by the fact that playing the optimal policy guarantees small violations independently on the episode the optimum is chosen. This is *not* the case of the stronger baseline, since playing the optimum in some episodes may lead to arbitrarily large violations.

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

## APPENDIX

The Appendix is structured as follows:

- In Section A, we provide additional related works.

- In Section B, we provide additional notation employed in the rest of the appendix.

- In Section C, we provide the events dictionary.

- In Section D we provide the theoretical guarantees attained by Algorithm 3. Precisely, we provide the complete version of the primal algorithm (see Algorithm 4), and analyze the related performances.

- In Section E, we provide the theoretical guarantees attained by the dual algorithm.

- In Section F, we provide the analysis to bound the Lagrange multipliers during the learning dynamic.

- In Section G, we provide the theoretical guarantees attained by Algorithm 2 when the constraints are stochastic.

- In Section H, we provide the theoretical guarantees attained by Algorithm 2 when the constraints are adversarial.

- In Section I we provide the theoretical guarantees attained by Algorithm 2 when the constraints are adversarial and the baseline is computed w.r.t. the policies that satisfy the constraints at each episode.

- In Section J we provide technical lemmas employed in our work.

- In Section K, we provide auxiliary lemmas from existing works.

## A   RELATED WORKS

In this section we provide further discussions on the works closely related to ours. We first provide some works in the field of unconstrained online MDPs (see (Auer et al., 2008; Even-Dar et al., 2009; Neu et al., 2010) for some initial results on the topic). The setting studied in these works generally differentiates the problem based on the nature of the losses (either stochastic or adversarial), the knowledge of the transition probability, and the nature of the feedback. Usually two types of feedback are considered: in the *full-information feedback* model, the entire loss function is observed after the learner's choice, while in the *bandit feedback* model, the learner only observes the loss due to the chosen action.

Azar et al. (2017) study the problem of optimal exploration in episodic MDPs with unknown transitions, stochastic losses and bandit feedback . The authors improve the previous result by Auer et al. (2008) designing an algorithm whose upper bound on the regret match the lower bound for this class, $\tilde{\mathcal{O}}(\sqrt{T})$. Rosenberg & Mansour (2019b) studies the setting of episodic MDPs with adversarial losses, unknown transitions, and full information feedback. In this case the authors present an online algorithm exploiting entropic regularization and providing a regret upper bound of $\tilde{\mathcal{O}}(\sqrt{T})$. The same setting is investigated when the feedback is bandit by Rosenberg & Mansour (2019a) who attain a regret upper bound of the order of $\tilde{\mathcal{O}}(T^{3/4})$, which is improved by Jin et al. (2020) by providing an algorithm that achieves in the same setting a regret upper bound of $\tilde{\mathcal{O}}(\sqrt{T})$. Finally, Luo et al. (2021) provide an optimal policy optimization algorithm for adversarial MDPs with bandit feedback.

In case of constrained problem, an fundamental result is presented by Mannor et al. (2009), who show that it is impossible to attain both sublinear regret and constraints violations when both the losses and constraints are adversarial. To overcome such an impossibility result, Liakopoulos et al. (2019) study a class of online learning problems with long-term budget constraints that can be chosen by an adversary and they define a new notion of regret. The new learner's regret metric introduces the notion of a *K-benchmark*, *i.e.*, a comparator that meets the problem's allotted budget over any window of length $K$. Castiglioni et al. (2022a;b) are the first to provide a *best-of-both-worlds* algorithm for online learning problems with long-term constraints, being the constraints stochastic or chosen by an adversary.

Constrained problems have been also studied in the context of CMDPs; however almost all previous works focus on the setting where the constraints are chosen stochastically. Wei et al. (2018) study the case of episodic CMDPs with known transition proabability, full-feedback, adversarial losses and stochastic constraints. The algorithm presented by the authors attains an upper bound both for constraints violation and for the regret of the order of $\tilde{\mathcal{O}}(\sqrt{T})$. Zheng & Ratliff (2020) present, in the setting of stochastic losses and constraints, where the transition probabilities are known and the feedback is bandit, an upper bound on the regret of their algorithm of the order of $\tilde{\mathcal{O}}(T^{3/4})$, while the cumulative constraint violations is guaranteed to be below a threshold with a given probability. Bai et al. (2020) provide the first algorithm to achieve sublinear regret when the transition probabilities are unknown, assuming that the rewards are deterministic and the constraints are stochastic with a particular structure. Efroni et al. (2020) study the case where transition probabilities, rewards, and constraints are unknown and stochastic, while the feedback is bandit. The authors propose two approaches to deal with the exploration-exploitation dilemma in episodic CMDPs guaranteeing sublinear regret and constraint violations. Qiu et al. (2020) provide a primal-dual approach based on *optimism in the face of uncertainty*. This work shows the effectiveness of such an approach when dealing with episodic CMDPs with adversarial losses and stochastic constraints, achieving both sublinear regret and constraint violation with full-information feedback. Wei et al. (2023), Ding & Lavaei (2023) and Stradi et al. (2024c) consider the case in which rewards and constraints are non-stationary, assuming that their variation is bounded. Thus, their results are *not* applicable to general adversarial settings. Stradi et al. (2024a), in the setting with adversarial losses, stochastic constraints and partial feedback, achieve sublinear regret and sublinear positive constraints violations. Finally, Stradi et al. (2024b) propose the first *best-of-both-worlds* algorithm for CMDPs, assuming *full feedback* on the rewards and constraints.

## B    ADDITIONAL NOTATION

In the following section, we introduce some useful notation from policy optimization. First, we define the value function $V^\pi(x; f)$, for policy $\pi$, state $x$ and generic function $f$ that assumes values for each state $x \in X$ and for each action $a \in A$. Formally,

$$V^\pi(x; f) := \mathbb{E}\left[\sum_{j=h(x)+1}^{H} f(x_j, a_j) | a_j \sim \pi(\cdot|x_j), x_j \sim P(\cdot|x_{j-1}, a_{j-1})\right],$$

where $h(x)$ is the layer $h$ such that $x \in X_h$. Notice that the value function can be written using the occupancy measure $q^{\pi,P}$ generated by the policy $\pi$ and the transition probability $P$ as : $V^\pi(x_0; f) = \sum_{x,a} q^{\pi,P}(x,a) f(x,a)$. We introduce also a $Q$-function of a generic function $f$ as:

$$\begin{cases} Q(x, a; f) = f(x, a) + \mathbb{E}_{x' \sim P(\cdot|x,a)}\left[V^\pi(x'; f)\right] \\ V^\pi(x; f) = \mathbb{E}_{a \sim \pi(\cdot|x)}\left[Q^\pi(x, a; f)\right] \\ V^\pi(x_H; f) = 0 \end{cases}$$

In addition we will use the notation $Q_t(x, a)$ to indicate the $Q$-function computed with respect to the function $\ell_t$, *i.e.* $Q(x, a; \ell_t)$.

## C    DICTIONARY

In the following, we provide the definition of different quantities which will be employed in the rest of the appendix. This is done for the ease of presentation.

- **Quantity $\mathcal{E}^P_{t_1,t_2}$:**

$$\mathcal{E}^P_{t_1,t_2} = U_1 \Xi_{t_1,t_2} C\sqrt{T} + U_2 \Xi_{t_1,t_2} \frac{(t_2 - t_1 + 1)}{C\sqrt{T}} + U_3 \Xi_{t_1,t_2} \frac{1}{C\sqrt{T}} + U_4 \Xi_{t_1,t_2} \sqrt{T},$$

  where:

  - $U_1 = 6H^2 \ln\left(\frac{H|A|T^2}{\delta}\right)$
  - $U_2 = 9H|X||A|$

– $U_3 = \frac{H}{2} \ln \left( \frac{HT^2}{\delta} \right)$

– $U_4 = 30H^2 |X|^2 \sqrt{2|A| \ln \left( \frac{T|X|^2|A|}{\delta} \right)}$.

With probability at least $1 - 4\delta$ it holds $R^P_{t_1,t_2} \leq \mathcal{E}^P_{t_1,t_2}$, $\forall t_1, t_2 \in [T] : 1 \leq t_1 \leq t_2 \leq T$ by Theorem 3.

• **Quantity $\mathcal{E}^D(\underline{0})$:**

$$\mathcal{E}^D(\underline{0}) = D_1 \frac{\|\lambda_{t_1}\|_2^2}{\eta} + D_2 \eta (t_2 - t_1 + 1),$$

where:

– $D_1 = \frac{1}{2}$

– $D_2 = \frac{mH^2}{2}$.

It holds $R^D_{t_1,t_2}(\underline{0}) \leq \mathcal{E}^D(\underline{0})$, $\forall t_1, t_2 \in [T] : 1 \leq t_1 \leq t_2 \leq T$ by Theorem 11.

• **Quantity $\mathcal{E}^G_{t_1,t_2}$:**

$$\mathcal{E}^G_{t_1,t_2} = B_1 \sqrt{(t_2 - t_1 + 1)},$$

where:

– $B_1 = 2H \sqrt{\ln \left( \frac{T^2}{\delta} \right)}$.

Given a $q \in \Delta(M)$, with probability at least $1 - \delta$ it holds in case of stochastic constraints $\sum_{t=t_1}^{t_2} (G_t^\top q - \overline{G}^\top q) \leq \mathcal{E}^G_{t_1,t_2}$, by Azuma-Hoeffding inequality.

• **Quantity $\mathcal{E}^{\mathbb{I}}_{t_1,t_2}$:**

$$\mathcal{E}^{\mathbb{I}}_{t_1,t_2} = F_1 \sqrt{(t_2 - t_1 + 1)},$$

where:

– $F_1 = H \sqrt{2 \ln \left( \frac{T^2}{\delta} \right)}$.

With probability at least $1 - \delta$ it holds $\sum_{t=t_1}^{t_2} \sum_{x,a} (\mathbb{I}_t(x,a) - q_t(x,a)) \leq \mathcal{E}^{\mathbb{I}}_{t_1,t_2}$, and with probability at least $1 - \delta$ it holds $\sum_{t=t_1}^{t_2} \sum_{x,a} (q_t(x,a) - \mathbb{I}_t(x,a)) \leq \mathcal{E}^{\mathbb{I}}_{t_1,t_2}$, by Azuma-Hoeffding inequality.

• **Quantity $C$:**

$$C = 252|X||A|H$$

• **Quantity $D$:**

$$D = 84672mH^2|X|^2|A|$$
$$= 336mH|X|C.$$

## D  OMITTED PROOFS FOR THE PRIMAL ALGORITHM

In this section we study the guarantees attained by the primal procedure, namely Algorithm 4.

---

**Algorithm 4** FS-PODB

---

**Require:** $X, A, \sigma = \frac{1}{T}, C$
1: $\mathcal{P}_1 \leftarrow$ set of all possible transitions
2: $\pi_1(a|x) = \frac{1}{|A|} \quad \forall(x,a) \in X \times A$
3: $\Xi_0 \leftarrow 1$
4: $\gamma \leftarrow \frac{1}{C\sqrt{T}}$
5: **for** $t = 1, \ldots, T$ **do**
6:      Play $\pi_t$, observe $\{(x_h, a_h)\}_{h=0}^{H-1}$, losses $\{\ell_t(x_h, a_h)\}_{h=0}^{H-1}$ and $\Xi_t$
7:      $\eta_t \leftarrow \frac{1}{2H\Xi_t C\sqrt{T}}$
8:      For all $h = 0, \ldots, H-1$ and $(x,a) \in X_h \times A$:

$$L_{t,h} = \sum_{j=h}^{H-1} \ell_t(x_h, a_h)$$

$$\widehat{Q}_t(x,a) = \frac{L_{t,h}}{\overline{q}_t(x,a) + \gamma} \mathbb{I}_t(x,a),$$

where $\overline{q}_t(x,a) = \max_{\widehat{P} \in \mathcal{P}_t} q^{\widehat{P}, \pi_t}(x,a)$ and $\mathbb{I}_t(x,a) = \mathbb{I}\{x_{t,h} = x, a_{t,h} = a\}$.

9:      For all $(x,a) \in X \times A$:

$$b_t(x) = \mathbb{E}_{a \sim \pi_t(\cdot|x)} \left[ \frac{3\gamma H\Xi_t + H\Xi_t \left( \overline{q}_t(x,a) - \underline{q}_t(x,a) \right)}{\overline{q}_t(x,a) + \gamma} \right]$$

$$B_t(x,a) = b_t(x) + \left(1 + \frac{1}{H}\right) \max_{\widehat{P} \in \mathcal{P}_t} \mathbb{E}_{x' \sim \widehat{P}(\cdot|x,a)} \mathbb{E}_{a \sim \pi_t(\cdot|x')} [B_t(x', a')]$$

where $\underline{q}_t = \min_{\widehat{P} \in \mathcal{P}_t} q^{\widehat{P}, \pi_t}(x,a)$, and $B_t(x_H, a) = 0$ for all $a$.

10:      For all $(x,a) \in X \times A$:

$$w_{t+1}(a|x) = (1-\sigma)w_t(a|x)e^{-\eta_t \left(\widehat{Q}_t(x,a) - B_t(x,a)\right)} + \frac{\sigma}{|A|} \sum_{a' \in A} w_t(a'|x)e^{-\eta_t \left(\widehat{Q}_t(x,a') - B_t(x,a')\right)}.$$

$$\pi_{t+1}(a|x) = \frac{w_{t+1}(a|x)}{\sum_{a' \in A} w_{t+1}(a'|x)}.$$

11:      $\mathcal{P}_{t+1} \leftarrow \texttt{TRANSITION.UPDATE}(\{(x_h, a_h)\}_{h=0}^{H-1})$

---

**Theorem 10.** *For any $\delta \in (0,1)$, Algorithm 4 attains, with probability at least $1 - 4\delta$ and for all $[t_1, \ldots, t_2] \subset [T]$:*

$$\sum_x q^*(x) \sum_{t=t_1}^{t_2} \sum_a (\pi_t(a|x) - \pi^*(a|x)) (Q_t^{\pi_t}(x,a) - B_t(x,a))$$

$$= \Xi_{t_1, t_2} o(T) + \sum_{t=t_1}^{t_2} V^{\pi^*}(x_0; b_t) + \frac{1}{H} \sum_{t=t_1}^{t_2} \sum_{x,a} q^*(x)\pi_t(a|x)B_t(x,a),$$

*for all $t_1, t_2 \in [T]$ s.t. $1 \le t_1 \le t_2 \le T$ and where $\Xi_{t_1, t_2} \ge \max_{t \in [t_1, \ldots, t_2]} \max_{x,a} \ell_t(x,a)$.*

*Proof.* In the rest of the proof, we will refer as $\bar{L}_t$ to $\max_{\tau \in [t]} \max_{h \in [H]} L_{\tau,h}$ and $\bar{L}_{t_1,t_2}$ to $\max_{\tau \in [t_1,\ldots,t_2]} \max_{h \in [H]} L_{\tau,h}$; therefore, by definition it holds $\bar{L}_t \le H\Xi_t$ for all $t \in [T]$.

As a first step, we decompose $\sum_x q^*(x) \sum_{t=t_1}^{t_2} \sum_a (\pi_t(a|x) - \pi^*(a|x)) (Q_t^{\pi_t}(x,a) - B_t(x,a))$ in three different quantities:

$$\sum_x q^*(x) \sum_{t=t_1}^{t_2} \sum_a (\pi_t(a|x) - \pi^*(a|x)) (Q_t^{\pi_t}(x,a) - B_t(x,a))$$

$$= \underbrace{\sum_x q^*(x) \sum_{t=t_1}^{t_2} \sum_a \left( \pi_t(a|x) - \pi^*(a|x) \right) \left( \widehat{Q}_t(x,a) - B_t(x,a) \right)}_{①}$$

$$+ \underbrace{\sum_x q^*(x) \sum_{t=t_1}^{t_2} \sum_a \pi_t(a|x) \left( Q_t^{\pi_t}(x,a) - \widehat{Q}_t(x,a) \right)}_{②}$$

$$+ \underbrace{\sum_x q^*(x) \sum_{t=t_1}^{t_2} \sum_a \pi^*(a|x) \left( \widehat{Q}_t(x,a) - Q_t^{\pi_t}(x,a) \right)}_{③},$$

which we proceed to bound separately.

**Bound on ①.** The quantity of interest can be bounded after noticing that Algorithm 4 employs a slightly modified version of OMD. In fact, recalling the definition of $\pi_t$, we can write:

$$\pi_{t+1}(a|x) = \frac{w_{t+1}(a|x)}{\sum_{a'} w_{t+1}(a'|x)}$$

$$= \frac{(1-\sigma)w_t(a|x)e^{-\eta_t\left(\widehat{Q}_t(x,a)-B_t(x,a)\right)} + \frac{\sigma}{|A|}\sum_{a'\in A} w_t(a'|x)e^{-\eta_t\left(\widehat{Q}_t(x,a')-B_t(x,a')\right)}}{\sum_{a'\in A} w_t(a'|x)e^{-\eta_t\left(\widehat{Q}_t(x,a')-B_t(x,a')\right)}}$$

$$= (1-\sigma)\frac{\pi_t(a|x)e^{-\eta\left(\widehat{Q}_t(x,a)-B_t(x,a)\right)}}{\sum_{a'} \pi_t(a'|x)e^{-\eta\left(\widehat{Q}_t(x,a')-B_t(x,a')\right)}} + \sigma\frac{1}{|A|}.$$

From now on we will refer to $\frac{\pi_t(a|x)e^{-\eta\left(\widehat{Q}_t(x,a)-B_t(x,a)\right)}}{\sum_{a'} \pi_t(a'|x)e^{-\eta\left(\widehat{Q}_t(x,a')-B_t(x,a')\right)}}$ as $\widetilde{\pi}_{t+1}(x,a)$. Thus,

$$\pi_{t+1}(a|x) = (1-\sigma)\widetilde{\pi}_{t+1}(x,a) + \frac{\sigma}{|A|}.$$

Calling $\psi(\cdot)$ the negative entropy function defined as $\psi(\pi(\cdot|x)) := \sum_a \pi(a|x) \ln(\pi(a|x))$, by standard analysis (*e.g.* Orabona (2019)), it holds:

$$\widetilde{\pi}_{t+1}(\cdot|x) = \operatorname*{arg\,min}_{\pi(\cdot|x)\in\Delta(A)} \sum_a \left( \widehat{Q}_t(x,a) - B_t(x,a) \right) \pi(a|x) + \frac{1}{\eta} D_\psi(\pi(\cdot|x); \pi_t(\cdot|x)),$$

where $D_\psi$ is Bregman divergence w.r.t. the negative entropy function $\psi(\cdot)$. Thus, for all $\pi(\cdot|x)$ it holds $\eta_t \sum_a \left( \widehat{Q}_t(x,a) - B_t(x,a) \right) (\pi(a|x) - \widetilde{\pi}_{t+1}(x,a)) + \langle \nabla\psi(\widetilde{\pi}_{t+1}(\cdot|x)) - \nabla\psi(\pi_t(\cdot|x)), \pi(\cdot|x) - \widetilde{\pi}_{t+1}(\cdot|x)\rangle \geq 0$. So, for all $\pi(\cdot|x)$ the following holds:

$$\eta_t\langle \widehat{Q}_t(x,\cdot) - B_t(x,\cdot), \pi_t(\cdot|x) - \pi(\cdot|x)\rangle$$

$$= \eta_t\langle \widehat{Q}_t(x,\cdot) - B_t(x,\cdot) + \nabla\psi(\widetilde{\pi}_{t+1}(\cdot|x)) - \nabla\psi(\pi_t(\cdot|x)), \widetilde{\pi}_{t+1}(\cdot|x) - \pi(\cdot|x)\rangle$$

$$+ \eta_t\langle \widehat{Q}_t(x,\cdot) - B_t(x,\cdot), \pi_t(\cdot|x) - \widetilde{\pi}_{t+1}(\cdot|x)\rangle$$

$$+ \langle \nabla\psi(\widetilde{\pi}_{t+1}(\cdot|x)) - \nabla\psi(\pi_t(\cdot|x)), \pi(\cdot|x) - \widetilde{\pi}_{t+1}(\cdot|x)\rangle$$

$$\leq \langle \eta_t\left( \widehat{Q}_t(x,\cdot) - B_t(x,\cdot)\right), \pi_t(\cdot|x) - \widetilde{\pi}_{t+1}(\cdot|x)\rangle$$

$$+ \langle \nabla\psi(\widetilde{\pi}_{t+1}(\cdot|x)) - \nabla\psi(\pi_t(\cdot|x)), \pi(\cdot|x) - \widetilde{\pi}_{t+1}(\cdot|x)\rangle$$

$$\leq D_\psi(\pi(\cdot|x); \pi_t(\cdot|x)) - D_\psi(\pi(\cdot|x); \widetilde{\pi}_{t+1}(\cdot|x)) - D_\psi(\widetilde{\pi}_{t+1}(\cdot|x); \pi_t(\cdot|x))$$

$$+ \eta_t\langle \widehat{Q}_t(x,\cdot) - B_t(x,\cdot), \pi_t(\cdot|x) - \widetilde{\pi}_{t+1}(\cdot|x)\rangle \qquad (3)$$

$$= D_\psi(\pi(\cdot|x); \pi_t(\cdot|x)) - D_\psi(\pi(\cdot|x); \widetilde{\pi}_{t+1}(\cdot|x)) + \frac{\eta_t^2}{2}\sum_{a\in A} \left( \widehat{Q}_t(x,a) - B_t(x,a) \right)^2 \pi_t(a|x), \quad (4)$$

where Inequality (3) and Inequality (4) are based on the proofs of Lemma 6.6. and Lemma 6.9. in Orabona (2019).

Additionally we can show that for all $t \in [T]$: $D_\psi(\pi(\cdot|x); \pi_t(\cdot|x)) - D_\psi(\pi(\cdot|x); \widetilde{\pi}_t(\cdot|x)) \leq \sigma \ln(|A|)$. Indeed,

$$D_\psi\left(\pi(\cdot|x); \pi_t(\cdot|x)\right) - D_\psi\left(\pi(\cdot|x); \widetilde{\pi}_t(\cdot|x)\right)$$

$$= D_\psi\left(\pi(\cdot|x); (1-\sigma)\widetilde{\pi}_t(\cdot|x) + \sigma\pi^{\frac{1}{|A|}}\right) - D_\psi\left(\pi(\cdot|x); \widetilde{\pi}_t(\cdot|x)\right)$$

$$\leq \sigma D_\psi\left(\pi(\cdot|x); \pi^{\frac{1}{|A|}}\right) - \sigma D_\psi\left(\pi(\cdot|x); \widetilde{\pi}_t(\cdot|x)\right)$$

$$\leq \sigma \ln(|A|),$$

where the last inequality holds since $D_\psi(\pi(\cdot|x); \widetilde{\pi}_t(\cdot|x)) \geq 0$ and

$$D_\psi(\pi(\cdot|x); \pi^{\frac{1}{|A|}}) = \sum_{a \in A} \pi(a|x) \ln\left(\frac{\pi(a|x)}{\pi^{\frac{1}{|A|}}(a|x)}\right)$$

$$\leq \sum_{a \in A} \pi(a|x) \ln\left(\frac{1}{\pi^{\frac{1}{|A|}}(a|x)}\right)$$

$$= \sum_{a \in A} \pi(a|x) \ln(|A|)$$

$$= \ln(|A|).$$

Notice that with we refer as $\pi^{\frac{1}{|A|}}$ to the vector strategy in $[0,1]^{|A|}$ with all elements equal to $\frac{1}{|A|}$.

Moreover we bound $D_\psi(\pi(\cdot|x); \pi_{t_1}(\cdot|x))$, since $\pi_{t_1}(a|x) = (1-\sigma)\widetilde{\pi}_{t_1}(a|x) + \sigma(\frac{1}{|A|}) \geq \frac{\sigma}{|A|}$, as follows:

$$D_\psi(\pi(\cdot|x); \pi_{t_1}(\cdot|x)) = \sum_{a \in A} \pi(a|x) \ln\left(\frac{\pi(a|x)}{\pi_{t_1}(a|x)}\right)$$

$$\leq \sum_{a \in A} \pi(a|x) \ln\left(\frac{1}{\pi_{t_1}(a|x)}\right)$$

$$\leq \sum_{a \in A} \pi(a|x) \ln\left(\frac{|A|}{\sigma}\right)$$

$$= \ln\left(\frac{|A|}{\sigma}\right).$$

Putting everything together we have that:

$$① = \sum_x q^*(x) \sum_{t=t_1}^{t_2} \sum_a (\pi_t(a|x) - \pi^*(a|x)) \left(\widehat{Q}_t(x,a) - B_t(x,a)\right)$$

$$\leq \sum_x q^*(x) \left(\frac{D_\psi(\pi(\cdot|x); \pi_{t_1}(\cdot|x))}{\eta_{t_1}} + \sum_{t=t_1+1}^{t_2} \left(\frac{D_\psi(\pi(\cdot|x); \pi_t(\cdot|x))}{\eta_t} - \frac{D_\psi(\pi(\cdot|x); \widetilde{\pi}_t(\cdot|x))}{\eta_{t+1}}\right)\right)$$

$$+ \sum_x q^*(x) \sum_{t=t_1}^{t_2} \frac{\eta_t}{2} \left(\widehat{Q}_t(x,a) - B_t(x,a)\right)^2 \pi_t(a|x) \quad (5a)$$

$$\leq \sum_x q^*(x) \left(\frac{D_\psi(\pi(\cdot|x); \pi_{t_1}(\cdot|x))}{\eta_{t_1}} + \sum_{t=t_1+1}^{t_2} \left(\frac{D_\psi(\pi(\cdot|x); \pi_t(\cdot|x)) - D_\psi(\pi(\cdot|x); \widetilde{\pi}_t(\cdot|x))}{\eta_t}\right)\right)$$

$$+ \sum_x q^*(x) \sum_{t=t_1}^{t_2} \frac{\eta_t}{2} \left(\widehat{Q}_t(x,a) - B_t(x,a)\right)^2 \pi_t(a|x) \quad (5b)$$

$$\leq \frac{\ln\left(\frac{|A|}{\sigma}\right)}{\eta_{t_1}} + \sigma \sum_{t=t_1+1}^{t_2} \frac{\ln(|A|)}{\eta_{t_2}} + \sum_x q^*(x) \sum_{t=t_1}^{t_2} \frac{\eta_t}{2} \left(\widehat{Q}_t(x,a) - B_t(x,a)\right)^2 \pi_t(a|x) \quad (5c)$$

$$\leq \frac{\ln\left(\frac{|A|}{\sigma}\right)}{\eta_{t_1}} + \frac{\sigma T \ln(|A|)}{\eta_{t_2}} + \sum_x q^*(x) \sum_{t=t_1}^{t_2} \frac{\eta_t}{2} \left(\widehat{Q}_t(x,a) - B_t(x,a)\right)^2 \pi_t(a|x)$$

$$= \frac{\ln(|A|T)}{\eta_{t_1}} + \frac{\ln(|A|)}{\eta_{t_2}} + \sum_x q^*(x) \sum_{t=t_1}^{t_2} \frac{\eta_t}{2} \left(\widehat{Q}_t(x,a) - B_t(x,a)\right)^2 \pi_t(a|x),$$

where $\sigma = \frac{1}{T}$, Inequality (5a) holds by Inequality (4) , Inequality (5b) holds since $\frac{1}{\eta_{t+1}} \geq \frac{1}{\eta_t}$ for all $t \in [T]$, and Inequality (5c) holds since $\eta_{t_2} \leq \eta_t$ for all $t$ in $[t_1+1, \ldots, t_2]$. Focusing now on the last part of the right term, with probability at least $1 - 2\delta$ the following holds:

$$\sum_x \sum_a q^*(x) \sum_{t=t_1}^{t_2} \frac{\eta_t}{2} \left(\widehat{Q}_t(x,a) - B_t(x,a)\right)^2 \pi_t(a|x)$$

$$\leq \sum_{t=t_1}^{t_2} \eta_t \sum_x \sum_a q^*(x)\pi_t(a|x)\widehat{Q}_t(x,a)^2 + \sum_{t=t_1}^{t_2} \eta_t \sum_x \sum_a q^*(x)\pi_t(a|x)B_t(x,a)^2 \qquad \text{(6a)}$$

$$= \sum_{t=t_1}^{t_2} \eta_t \sum_x \sum_a q^*(x)\pi_t(a|x)\frac{L_{t,h}^2}{(\overline{q}_t(x,a)+\gamma)^2}\mathbb{I}_t(x,a) + \sum_{t=t_1}^{t_2} \eta_t \sum_x \sum_a q^*(x)\pi_t(a|x)B_t(x,a)^2 \qquad \text{(6b)}$$

$$\leq \bar{L}_{t_1,t_2} \sum_{t=t_1}^{t_2} \eta_t \bar{L}_t \sum_x \sum_a \frac{q^*(x)\pi_t(a|x)}{\overline{q}_t(x,a)+\gamma}\frac{\mathbb{I}_t(x,a)}{\overline{q}_t(x,a)+\gamma} + \sum_{t=t_1}^{t_2} \eta_t \sum_x \sum_a q^*(x)\pi_t(a|x)B_t(x,a)^2 \qquad \text{(6c)}$$

$$\leq \frac{\gamma}{2H} \bar{L}_{t_1,t_2} \sum_{t=t_1}^{t_2} \sum_x \sum_a \frac{q^*(x)\pi_t(a|x)}{\overline{q}_t(x,a)+\gamma}\frac{q_t(x,a)}{\overline{q}_t(x,a)+\gamma} + \frac{\gamma\bar{L}_{t_1,t_2}}{2} \ln\left(\frac{HT^2}{\delta}\right)$$

$$+ \sum_{t=t_1}^{t_2} \eta_t \sum_x \sum_a q^*(x)\pi_t(a|x)B_t(x,a)^2 \qquad \text{(6d)}$$

$$\leq \sum_{t=t_1}^{t_2} \sum_x \sum_a q^*(x)\pi_t(a|x)\frac{\gamma\Xi_{t_1,t_2}}{2(\overline{q}_t(x,a)+\gamma)} + \frac{\gamma\bar{L}_{t_1,t_2}}{2} \ln\left(\frac{HT^2}{\delta}\right)$$

$$+ \frac{1}{2H} \sum_{t=t_1}^{t_2} \sum_x \sum_a q^*(x)\pi_t(a|x)B_t(x,a) \qquad \text{(6e)}$$

$$= \sum_{t=t_1}^{t_2} \sum_x \sum_a q^*(x)\pi_t(a|x)\left(\frac{\gamma\Xi_{t_1,t_2}}{2(\overline{q}_t(x,a)+\gamma)} + \frac{B_t(x,a)}{2H}\right) + \frac{\gamma\bar{L}_{t_1,t_2}}{2} \ln\left(\frac{HT^2}{\delta}\right),$$

where Inequality (6a) holds since $(a-b)^2 \leq 2a^2 + 2b^2$, for all $a, b \in \mathbb{R}$, Equality (6b) holds by definition of $\widehat{Q}(x,a)$, Inequality (6c) is motivated by the fact that $L_{t,h} \leq \bar{L}_{t_1,t_2}$ by its definition, Inequality (6d) holds with probability at least $1 - \delta$ by applying Lemma 9 and taking $\alpha_t(x,a) = \frac{q^*(x)\pi_t(a|x)}{\overline{q}_t(x,a)+\gamma}$ since $\frac{q^*(x)\pi_t(a|x)}{\overline{q}_t(x,a)+\gamma} \leq \frac{1}{\gamma}$ and considering that by definition $\eta_t\Xi_t = \frac{\gamma}{2H}$, and finally Inequality (6e) holds since $\overline{q}_t(x,a) \geq q_t(x,a)$, $\forall(x,a) \in X \times A, \forall t \in [t_1 \ldots t_2]$ with probability at least $1 - \delta$ by Lemma 12 and by Lemma 6. Setting $\gamma = 2\eta_t H\Xi_t$, we can conclude that, with probability at least $1 - 2\delta$, ① is bounded as:

$$\frac{4H\Xi_{t_1,t_2}\ln(|A|T)}{\gamma} + \gamma\frac{H\Xi_{t_1,t_2}}{2} \ln\left(\frac{HT^2}{\delta}\right) + \sum_{t=t_1}^{t_2} \sum_x \sum_a q^*(x)\pi_t(a|x)\left(\frac{\gamma\Xi_{t_1,t_2}}{2(\overline{q}_t(x,a)+\gamma)} + \frac{B_t(x,a)}{2H}\right).$$

**Bound on** ②. To bound ② we employ the same approach as in (Luo et al., 2021). First we define $Y_t$ as $Y_t := \sum_x \sum_a q^*(x)\pi_t(a|x)\widehat{Q}_t(x,a)$, for all $t \in [T]$. Now since $\sum_{t=t_1}^{t_2} Y_t$ is a martingale sequence , we apply Freedman's inequality. First notice that under the event $P \in \mathcal{P}_i(t)$ for all

$t \in [T]$:

$$
\mathbb{E}[Y_t^2] \leq \mathbb{E}_t\left[\left(\sum_x \sum_a q^*(x)\pi_t(a|x)\widehat{Q}_t(x,a)\right)^2\right]
$$

$$
\leq \mathbb{E}_t\left[\left(\sum_x \sum_a q^*(x)\pi_t(a|x)\right)\left(\sum_x \sum_a q^*(x)\pi_t(a|x)\widehat{Q}_t(x,a)^2\right)\right]
$$

$$
= H\mathbb{E}_t\left[\sum_x \sum_a q^*(x)\pi_t(a|x)\widehat{Q}_t(x,a)^2\right]
$$

$$
= H\sum_x \sum_a q^*(x)\pi_t(a|x)\frac{L_{t,h}^2}{(\overline{q}_t(x,a)+\gamma)^2}q_t(x,a)
$$

$$
\leq \sum_x \sum_a q^*(x)\pi_t(a|x)\frac{H\bar{L}_t^2}{\overline{q}_t(x,a)+\gamma}.
$$

Thus, thanks to Lemma 8, since $|Y_t| \leq H\sup_{x',a'}\widehat{Q}_t(x,a) \leq \frac{H\bar{L}_t}{\gamma}$, with probability at least $1-\delta$ it holds simultaneously for all $t_1, t_2 : 1 \leq t_1 \leq t_2 \leq T$:

$$
\sum_{t=t_1}^{t_2}(\mathbb{E}_t[Y_t] - Y_t) \leq \frac{\gamma}{H\bar{L}_{t_1,t_2}}\sum_{t=t_1}^{t_2}\sum_x \sum_a q^*(x)\pi_t(a|x)\frac{H\bar{L}_t^2}{\overline{q}_t(x,a)+\gamma} + \frac{H\bar{L}_{t_1,t_2}}{\gamma}\log\left(\frac{T^2}{\delta}\right).
$$

We notice also the following result with probability at least $1-\delta$ for all $t \in [T]$:

$$
\sum_x \sum_a q^*(x)\pi_t(a|x)Q_t(x,a) - \mathbb{E}[Y_t]
$$

$$
= \sum_x \sum_a q^*(x)\pi_t(a|x)Q_t(x,a) - \mathbb{E}\left[\sum_x \sum_a q^*(x)\pi_t(a|x)\widehat{Q}_t(x,a)\right]
$$

$$
= \sum_x \sum_a q^*(x)\pi_t(a|x)Q_t(x,a)\left(1 - \frac{q_t(x,a)}{\overline{q}_t(x,a)+\gamma}\right)
$$

$$
\leq \sum_x \sum_a q^*(x)\pi_t(a|x)H\Xi_t\left(\frac{\overline{q}_t(x,a) - q_t(x,a) + \gamma}{\overline{q}_t(x,a)+\gamma}\right)
$$

$$
\leq \sum_x \sum_a q^*(x)\pi_t(a|x)H\Xi_t\left(\frac{(\overline{q}_t(x,a) - \underline{q}_t(x,a)) + \gamma}{\overline{q}_t(x,a)+\gamma}\right).
$$

Finally we can bound ② with probability at least $1-2\delta$ as follows.

$$
② = \sum_x q^*(x)\sum_{t=t_1}^{t_2}\sum_a \pi_t(a|x)\left(Q_t^{\pi_t}(x,a) - \widehat{Q}_t(x,a)\right)
$$

$$
= \sum_{t=t_1}^{t_2}(\mathbb{E}_t[Y_t] - Y_t) + \sum_{t=t_1}^{t_2}\left(\sum_x \sum_a q^*(x)\pi_t(a|x)Q_t(x,a) - \mathbb{E}[Y_t]\right)
$$

$$
\leq \sum_{t=t_1}^{t_2}\sum_x \sum_a q^*(x)\pi_t(a|x)H\Xi_t\left(\frac{(\overline{q}_t(x,a) - \underline{q}_t(x,a)) + 2\gamma}{\overline{q}_t(x,a)+\gamma}\right) + \frac{H\bar{L}_{t_1,t_2}}{\gamma}\ln\left(\frac{T^2}{\delta}\right).
$$

**Bound on ③.** With probability at least $1-2\delta$ it holds:

$$
③ = \sum_x q^*(x)\sum_{t=t_1}^{t_2}\sum_a \pi^*(a|x)\left(\widehat{Q}_t(x,a) - Q_t^{\pi_t}(x,a)\right) \leq \frac{H^2\Xi_{t_1,t_2}}{2\gamma}\ln\left(\frac{HT^2}{\delta}\right),
$$

by Corollary 1.

**Conclusion of the proof** Finally we notice that, with probability at least $1 - 4\delta$, we have the following result.

$$\sum_x q^*(x) \sum_{t=t_1}^{t_2} \sum_a \left(\pi_t(a|x) - \pi^*(a|x)\right) \left(Q_t^{\pi_t}(x,a) - B_t(x,a)\right) = ① + ② + ③$$

$$\leq \gamma \frac{H\Xi_{t_1,t_2}}{2} \ln\left(\frac{HT^2}{\delta}\right) + \frac{6H^2\Xi_{t_1,t_2}}{\gamma} \ln\left(\frac{H|A|T^2}{\delta}\right)$$

$$+ \sum_{t=t_1}^{t_2} \sum_{x,a} q^*(x)\pi_t(a|x) \left(\frac{\Xi_t(3\gamma H + H(\overline{q}_t(x,a) - \underline{q}_t(x,a)))}{\overline{q}_t(x,a) + \gamma} + \frac{B_t(x,a)}{H}\right).$$

This concludes the proof. $\qquad\qquad\qquad\qquad\qquad\qquad\qquad\qquad\qquad\qquad\square$

**Theorem 3.** *For any $\delta \in (0,1)$, with probability at least $1 - 8\delta$, Algorithm* `FS-PODB` *attains:*

$$R_{t_1,t_2} \leq \widetilde{\mathcal{O}}\left(\Xi_{t_1,t_2}\sqrt{T} + \Xi_{t_1,t_2}\frac{t_2 - t_1}{\sqrt{T}}\right),$$

*where the regret can be computed with respect to any policy function $\pi : X \to \Delta(A)$.*

*Proof.* By means of Theorem 10 and by Lemma 7 we have that with probability at least $1 - 4\delta$:

$$R_{t_1,t_2}^P \leq \gamma \frac{H\Xi_{t_1,t_2}}{2} \ln\left(\frac{HT^2}{\delta}\right) + \frac{6H^2\Xi_{t_1,t_2}}{\gamma} \ln\left(\frac{H|A|T^2}{\delta}\right) + 3\sum_{t=t_1}^{t_2} \widehat{V}^{\pi_t}(x_0; b_t).$$

We can bound $\sum_{t=1}^T \widehat{V}^{\pi_t}(x_0; b_t)$, with probability at least $1 - 4\delta$, as:

$$\sum_{t=t_1}^{t_2} \widehat{V}^{\pi_t}(x_0; b_t)$$

$$= \sum_{t=t_1}^{t_2} \sum_{x,a} q^{\widehat{P}_t, \pi_t}(x,a) \left(\frac{H\Xi_t(\overline{q}_t(x,a) - \underline{q}_t(x,a)) + 3H\Xi_t\gamma}{\overline{q}_t(x,a) + \gamma}\right)$$

$$\leq \sum_{t=t_1}^{t_2} \sum_{x,a} H\Xi_t \left((\overline{q}_t(x,a) - \underline{q}_t(x,a)) + 3\gamma\right)$$

$$\leq \sum_{t=t_1}^{t_2} \sum_{x,a} H\Xi_t(\overline{q}_t(x,a) - \underline{q}_t(x,a)) + 3\Xi_{t_1,t_2}\gamma H(t_2 - t_1 + 1)|X||A|$$

$$\leq 4H^2\Xi_{t_1,t_2}|X|^2\sqrt{2T\ln\left(\frac{H|X|}{\delta}\right)} + 6\Xi_{t_1,t_2}H^2|X|^2\sqrt{2T|A|\ln\left(\frac{T|X|^2|A|}{\delta}\right)}$$

$$+ 3\Xi_{t_1,t_2}\gamma H|X||A|(t_2 - t_1 + 1),$$

where the second inequality holds under the event $q^{\widehat{P}_t, \pi_t}(x,a) \leq \overline{q}_t(x,a)$ for all $(x,a) \in X \times A$ and for all $t \in [T]$ and the last inequality uses Lemma 10. Thus, with probability at least $1 - 8\delta$, it holds:

$$R_{t_1,t_2}^P$$

$$\leq \gamma \frac{H\Xi_{t_1,t_2}}{2} \ln\left(\frac{HT^2}{\delta}\right) + \frac{6H^2\Xi_{t_1,t_2}}{\gamma} \ln\left(\frac{H|A|T^2}{\delta}\right) + 30\Xi_{t_1,t_2}H^2|X|^2\sqrt{2T|A|\ln\left(\frac{T|X|^2|A|}{\delta}\right)} +$$

$$+ 9\Xi_{t_1,t_2}\gamma H|X||A|(t_2 - t_1 + 1)$$

$$\leq \frac{H\Xi_{t_1,t_2}}{2C\sqrt{T}} \ln\left(\frac{HT^2}{\delta}\right) + 6H^2\Xi_{t_1,t_2}C\sqrt{T}\ln\left(\frac{H|A|T^2}{\delta}\right)$$

$$+ 30\Xi_{t_1,t_2}H^2|X|^2\sqrt{2T|A|\ln\left(\frac{T|X|^2|A|}{\delta}\right)} + 9H|X||A|\Xi_{t_1,t_2}\frac{(t_2 - t_1 + 1)}{C\sqrt{T}}$$

$$= U_1 \Xi_{t_1,t_2} C\sqrt{T} + U_2 \Xi_{t_1,t_2} \frac{(t_2 - t_1 + 1)}{C\sqrt{T}} + U_3 \Xi_{t_1,t_2} \frac{1}{C\sqrt{T}} + U_4 \Xi_{t_1,t_2} \sqrt{T}$$

$$= \mathcal{E}^P_{t_1,t_2},$$

which concludes the proof. $\square$

# E  OMITTED PROOFS FOR THE DUAL ALGORITHM

**Theorem 11.** *When employed by Algorithm 2, online projected gradient descent (OGD) attains:*

$$R^D_{t_1,t_2}(\lambda) = \sum_{t=t_1}^{t_2} (\lambda - \lambda_t)^\top \sum_{h=0}^{H-1} G_t(x_h, a_h) \leq \frac{\|\lambda_{t_1} - \lambda\|_2^2}{2\eta} + \frac{\eta}{2}(t_2 - t_1 + 1)mH^2.$$

*Proof.* We proceed to prove the theorem following (Orabona, 2019). Indeed, it holds:

$$R^D_{t_1,t_2}(\lambda) = \sum_{t=t_1}^{t_2} (\lambda - \lambda_t)^\top \sum_{h=0}^{H-1} G_t(x_h, a_h)$$

$$\leq \frac{\|\lambda_{t_1} - \lambda\|_2^2}{2\eta} + \frac{\eta}{2} \sum_{t=t_1}^{t_2} \| \sum_{h=0}^{H-1} G_t(x_h, a_h)\|_2^2$$

$$\leq \frac{\|\lambda_{t_1} - \lambda\|_2^2}{2\eta} + \frac{\eta}{2} \sum_{t=t_1}^{t_2} \sum_{i=1}^{m} \left( \sum_{h=0}^{H-1} G_{t,i}(x_h, a_h) \right)^2$$

$$\leq \frac{\|\lambda_{t_1} - \lambda\|_2^2}{2\eta} + \frac{\eta}{2} \sum_{t=t_1}^{t_2} mH^2$$

$$\leq \frac{\|\lambda_{t_1} - \lambda\|_2^2}{2\eta} + \frac{\eta}{2}(t_2 - t_1 + 1)mH^2.$$

This concludes the proof. $\square$

**Lemma 1.** *When employed by Algorithm 2, online projected gradient descent (OGD) guarantees for all $t \in [T]$:*

$$\|\lambda_{t+1}\|_1 - \|\lambda_t\|_1 \leq mH\eta.$$

*Proof.* It holds:

$$\lambda_{t+1,i} = \min\left\{ \max\left\{ 0, \lambda_{t,i} + \eta \sum_{h=0}^{H-1} g_{t,i}(x_h, a_h) \right\}, T^{\frac{1}{4}} \right\}$$

$$\leq \max\left\{ 0, \lambda_{t,i} + \eta \sum_{h=0}^{H-1} g_{t,i}(x_h, a_h) \right\}$$

$$\leq \max\left\{ 0, \lambda_{t,i} + \eta \sum_{h=0}^{H-1} 1 \right\}$$

$$= \lambda_{t,i} + \eta H,$$

which concludes the proof when we take the sum over all $i \in [m]$. $\square$

# F  ANALYSIS ON LAGRANGIAN MULTIPLIERS

**Lemma 2.** *The loss given to the primal algorithm at episode $t \in [T]$, which is defined as $\ell_t(x, a) = \Gamma_t + \sum_{i\in[m]} \lambda_{t,i} g_{t,i}(x, a) - r_t(x, a)$, is such that, considering the Lagrangian loss function $\ell^{\mathcal{L}}_t(x, a) = \sum_{i\in[m]} \lambda_{t,i} g_{t,i}(x, a) - r_t(x, a)$, it holds:*

$$\ell_t^\top (q_t - q^*) = \ell_t^{\mathcal{L},\top}(q_t - q^*),$$

*and additionally, $\ell_t$ assume values in the bounded interval $[0, \Xi_t]$.*

*Proof.* By simple computation, it holds:

$$\ell_t^\top (q_t - q^*) - \ell_t^{\mathcal{L},\top}(q_t - q^*)$$

$$= \left( \sum_{x,a} \Gamma_t(q_t(x,a) - q^*(x,a)) + \sum_{i \in [m]} \sum_{x,a} \lambda_{t,i} g_{t,i}(x,a)(q_t(x,a) - q^*(x,a)) \right.$$

$$\left. - \sum_{x,a} r_t(x,a)(q_t(x,a) - q^*(x,a)) \right)$$

$$- \left( \sum_{i \in [m]} \sum_{x,a} \lambda_{t,i} g_{t,i}(x,a)(q_t(x,a) - q^*(x,a)) - \sum_{x,a} r_t(x,a)(q_t(x,a) - q^*(x,a)) \right)$$

$$= \sum_{x,a} \Gamma_t(q_t(x,a) - q^*(x,a))$$

$$= \Gamma_t(H - H)$$

$$= 0,$$

where the last steps hold since $\Gamma_t$ is a constant and by the definition of *valid* occupancy measures.

In addition it holds:

$$\ell_t(x,a) = \Gamma_t + \sum_{i \in [m]} \lambda_{t,i} g_{t,i}(x,a) - r_t(x,a) \geq 1 + \|\lambda_t\|_1 - \sum_{i \in [m]} \lambda_{t,i} - 1 = 0,$$

and similarly,

$$\ell_t(x,a) = \Gamma_t + \sum_{i \in [m]} \lambda_{t,i} g_{t,i}(x,a) - r_t(x,a) \leq \Gamma_t + \sum_{i \in [m]} \lambda_{t,i} = 1 + 2\|\lambda_t\| \leq \Xi_t.$$

This concludes the proof. $\qquad\square$

**Theorem 4.** *Under Condition 2, for any $\delta \in (0,1)$, with probability at least $1 - 11\delta$, it holds:*

$$\|\lambda_t\|_1 \leq \Lambda \quad \forall t \in [T+1],$$

*where $\Lambda = \frac{112mH^2}{\rho^2}$.*

*Proof.* Let $M > 1$ be a constant. By absurd suppose $\exists t_2 \in [T]$ s.t.

$$\forall t \leq t_2 \quad \|\lambda_t\|_1 \leq \frac{2HM}{\rho^2} \quad \wedge \quad \|\lambda_{t_2+1}\|_1 > \frac{2HM}{\rho^2} \tag{7}$$

and let $t_1 < t_2$ s.t.

$$\|\lambda_{t_1-1}\|_1 \leq \frac{2H}{\rho} \quad \wedge \quad \forall t : t_1 \leq t \leq t_2 \|\lambda_t\|_1 \geq \frac{2H}{\rho}.$$

By construction $1 < \frac{2H}{\rho} \leq \|\lambda_t\|_1 \leq \frac{2HM}{\rho^2}$ for all $t_1 \leq t \leq t_2$, and it holds if $\eta \leq \frac{1}{mH}$:

$$\|\lambda_{t_1}\|_1 \leq \|\lambda_{t_1-1}\|_1 + m\eta H \leq \frac{2H}{\rho} + m\eta H \leq \frac{4H}{\rho}. \tag{8}$$

Notice also that by construction, calling $\lambda_{t_1,t_2} = \max_{t \in [t_1,\ldots t_2]} \|\lambda_t\|_1$, it holds:

$$1 < \lambda_{t_1,t_2} \leq \frac{2HM}{\rho^2} \quad \wedge \quad 1 + \lambda_{t_1,t_2} < \frac{4HM}{\rho^2}. \tag{9}$$

In the stochastic setting the following holds by Azuma-Hoeffding inequality with probability at least $1 - \delta$:

$$\sum_{t=t_1}^{t_2} -\lambda_t^\top G_t^\top q^\circ \geq \sum_{t=t_1}^{t_2} -\lambda_t^\top \overline{G}^\top q^\circ - \lambda_{t_1,t_2} \mathcal{E}_{t_1,t_2}^G$$

$$\geq \lambda_{t_1,t_2}(t_2 - t_1 + 1)\rho - \lambda_{t_1,t_2}\mathcal{E}^G_{t_1,t_2}$$
$$\geq (t_2 - t_1 + 1)2H - \lambda_{t_1,t_2}\mathcal{E}^G_{t_1,t_2},$$

where $\mathcal{E}^G_{t_1,t_2} = B_1\sqrt{(t_2 - t_1 + 1)} = 2H\sqrt{\ln\left(T^2/\delta\right)}\sqrt{(t_2 - t_1 + 1)}$. Instead, in the adversarial setting, it holds:

$$\sum_{t=t_1}^{t_2} -\lambda_t^\top G_t^\top q^\circ \geq \sum_{t=t_1}^{t_2}\sum_{i=1}^{m} -\lambda_{t,i}\left[G_t^\top q^\circ\right]_i$$

$$\geq \rho\sum_{t=t_1}^{t_2}\sum_{i=1}^{m}\lambda_{t,i}$$

$$= \rho\sum_{t=t_1}^{t_2}\|\lambda_t\|_1$$

$$\geq (t_2 - t_1 + 1)2H.$$

Generalizing the result, it holds, both for the stochastic and the adversarial setting, the following inequality with probability equal to 1 in the adversarial case and with probability at least $1 - \delta$ in the stochastic case:

$$\sum_{t=t_1}^{t_2} -\lambda_t^\top G_t^\top q^\circ \geq (t_2 - t_1 + 1)2H - \lambda_{t_1,t_2}\mathcal{E}^G_{t_1,t_2}.$$

Thank to this result we can find a lower bound for $-\sum_{t=t_1}^{t_2}\ell_t^{\mathcal{L},\top}q_t$ with probability at least $1 - 9\delta$ in the stochastic setting and with probability at least $1 - 8\delta$ in the adversarial case, employing Theorem 3 .

$$-\sum_{t=t_1}^{t_2}\ell_t^{\mathcal{L},\top}q_t = \sum_{t=t_1}^{t_2}(r_t^\top q^\circ - \lambda^\top G_t^\top q^\circ) - \sum_{t=t_1}^{t_2}\ell_t^{\mathcal{L},\top}(q_t - q^\circ)$$

$$\geq \sum_{t=t_1}^{t_2} -\lambda^\top G_t^\top q^\circ - \sum_{t=t_1}^{t_2}\ell_t^{\mathcal{L},\top}(q_t - q^\circ) \qquad (10)$$

$$\geq 2H(t_2 - t_1 + 1) - \lambda_{t_1,t_2}\mathcal{E}^G_{t_1,t_2} - \mathcal{E}^P_{t_1,t_2}, \qquad (11)$$

where Inequality (10) holds since $r_t^\top q^\circ \geq 0$, for all $t \in [T]$, and Inequality (11) is derived using the bound on the primal interval regret given by Theorem 3 and defined as $\mathcal{E}^P_{t_1,t_2}$ and by Lemma 2.

At the same time, it is possible to define also an upper bound for the same quantity $-\sum_{t=t_1}^{t_2}\ell_t^{\mathcal{L},\top}q_t$ with probability at least $1 - 2\delta$:

$$-\sum_{t=t_1}^{t_2}\ell_t^{\mathcal{L},\top}q_t = \sum_{t=t_1}^{t_2}(r_t^\top q_t - \lambda_t^\top G_t^\top q_t)$$

$$\leq \sum_{t=t_1}^{t_2}H - \sum_{t=t_1}^{t_2}\lambda_t^\top\left(G_t^\top q_t - \sum_{h=0}^{H-1}G_t(x_h, a_h)\right) + \sum_{t=t_1}^{t_2}(\underline{0} - \lambda_t)\sum_{h=0}^{H-1}G_t(x_h, a_h)$$

$$\leq H(t_2 - t_1 + 1) + \lambda_{t_1,t_2}\sum_{t=t_1}^{t_2}\sum_{x,a}G_t(x, a)(\mathbb{I}_t(x, a) - q_t(x, a)) + \mathcal{E}^D_{t_1,t_2}(\underline{0})$$

$$\leq H(t_2 - t_1 + 1) + \lambda_{t_1,t_2}\mathcal{E}^{\mathbb{I}}_{t_1,t_2} + \mathcal{E}^D_{t_1,t_2}(\underline{0}),$$

where $\mathcal{E}^{\mathbb{I}} = F_1\sqrt{(t_2 - t_1 + 1)} = H\sqrt{2\ln\left(T^2/\delta\right)}\sqrt{(t_2 - t_1 + 1)}$ and $\mathcal{E}^D(\underline{0}) = D_1\frac{\|\lambda_{t_1}\|_2^2}{\eta} + D_2\eta(t_2 - t_1 + 1) = \frac{1}{2}\frac{\|\lambda_{t_1}\|_2^2}{\eta} + \frac{mH^2}{2}\eta(t_2 - t_1 + 1)$. Thus, combining the two bounds we get with probability at least $1 - 10\delta$ in the adversarial case and $1 - 11\delta$ in the stochastic case the following bound,

$$2H(t_2 - t_1 + 1) - \lambda_{t_1,t_2}\mathcal{E}^G_{t_1,t_2} - \mathcal{E}^P_{t_1,t_2} \leq H(t_2 - t_1 + 1) + \lambda_{t_1,t_2}\mathcal{E}^{\mathbb{I}}_{t_1,t_2} + \mathcal{E}^D_{t_1,t_2}(\underline{0}),$$

which can be reordered as

$$H(t_2 - t_1 + 1) \leq \lambda_{t_1,t_2} \mathcal{E}^G_{t_1,t_2} + \lambda_{t_1,t_2} \mathcal{E}^{\mathbb{I}}_{t_1,t_2} + \mathcal{E}^D_{t_1,t_2}(\underline{0}) + \mathcal{E}^P_{t_1,t_2}.$$

We recall here the definitions of the bounds $\mathcal{E}^G_{t_1,t_2}, \mathcal{E}^{\mathbb{I}}_{t_1,t_2}, \mathcal{E}^D_{t_1,t_2}(\underline{0})$ and $\mathcal{E}^P_{t_1,t_2}$.

$$\mathcal{E}^G_{t_1,t_2} = B_1 \sqrt{(t_2 - t_1 + 1)},$$

where $B_1 = 2H\sqrt{\ln\left(\frac{T^2}{\delta}\right)}$.

$$\mathcal{E}^{\mathbb{I}}_{t_1,t_2} = F_1 \sqrt{(t_2 - t_1 + 1)},$$

where $F_1 = H\sqrt{2\ln\left(\frac{T^2}{\delta}\right)}$.

$$\mathcal{E}^D_{t_1,t_2}(\underline{0}) = D_1 \frac{\|\lambda_{t_1}\|_2^2}{\eta} + D_2\eta(t_2 - t_1 + 1),$$

where $D_1 = \frac{1}{2}$, $D_2 = \frac{mH^2}{2}$.

$$\mathcal{E}^P_{t_1,t_2} = U_1\Xi_{t_1,t_2}C\sqrt{T} + U_2\Xi_{t_1,t_2}\frac{(t_2 - t_1 + 1)}{C\sqrt{T}} + U_3\Xi_{t_1,t_2}\frac{1}{C\sqrt{T}} + U_4\Xi_{t_1,t_2}\sqrt{T},$$

where $U_1 = 6H^2\ln\left(\frac{H|A|T^2}{\delta}\right)$, $U_2 = 9H|X||A|$, $U_3 = \frac{H}{2}\ln\left(\frac{HT^2}{\delta}\right)$ and $U_4 = 30H^2|X|^2\sqrt{2|A|\ln\left(\frac{T|X|^2|A|}{\delta}\right)}$.

Thus, we can write:

$$H(t_2 - t_1 + 1) \leq \underbrace{\lambda_{t_1,t_2}(F_1 + B_1)\sqrt{(t_2 - t_1 + 1)}}_{\textcircled{1}} + \underbrace{U_2\Xi_{t_1,t_2}\frac{t_2 - t_1 + 1}{C\sqrt{T}}}_{\textcircled{2}}$$

$$+ \underbrace{U_3\Xi_{t_1,t_2}\frac{1}{C\sqrt{T}}}_{\textcircled{3}} + \underbrace{U_1\Xi_{t_1,t_2}C\sqrt{T}}_{\textcircled{4}} + \underbrace{D_2\eta(t_2 - t_1 + 1)}_{\textcircled{5}}$$

$$+ \underbrace{D_1\frac{\|\lambda_{t_1}\|_2^2}{\eta}}_{\textcircled{6}} + \underbrace{U_4\Xi_{t_1,t_2}\sqrt{T}}_{\textcircled{7}}.$$

To conclude the proof by absurd it is sufficient to prove that all $\textcircled{1}, \textcircled{2}, \textcircled{3}, \textcircled{4}, \textcircled{5}, \textcircled{6}, \textcircled{7}$ are smaller or equal to $\frac{H(t_2-t_1+1)}{7}$, with at least one being strictly smaller.

$\boxed{\text{Prove } \textcircled{1} < \frac{H(t_2-t_1+1)}{7}}$ If $\eta \leq \frac{1}{14m(F_1+B_1)\sqrt{T}}$, then $\textcircled{1} < \frac{H(t_2-t_1+1)}{7}$ holds. Indeed:

$$\frac{H(t_2 - t_1 + 1)}{7} > \frac{HM}{7\rho^2 m\eta} \tag{12a}$$

$$\geq \frac{2HM}{\rho^2}(F_1 + B_1)\sqrt{T} \tag{12b}$$

$$\geq \lambda_{t_1,t_2}(F_1 + B_1)\sqrt{T} \tag{12c}$$

$$\geq \lambda_{t_1,t_2}(F_1 + B_1)\sqrt{t_2 - t_1 + 1},$$

where Inequality (12a) holds by Lemma 11, Inequality (12b) is equivalent to condition $\eta \leq \frac{1}{14m(F_1+B_1)\sqrt{T}}$ and Inequality (12c) is true by Assumption (7).

$\boxed{\text{Prove } \textcircled{2} < \frac{H(t_2-t_1+1)}{7}}$ If $C \geq 56\frac{MU_2}{\rho^2\sqrt{T}}$ holds, then $\textcircled{2} < \frac{H(t_2-t_1+1)}{7}$ also holds. Indeed:

$$\frac{H(t_2 - t_1 + 1)}{7} \geq 2U_2\left(\frac{4HM}{\rho^2}\right)\frac{t_2 - t_1 + 1}{C\sqrt{T}} \tag{13a}$$

$$> U_2 2(1 + \lambda_{t_1,t_2}) \frac{t_2 - t_1 + 1}{C\sqrt{T}} \tag{13b}$$

$$\geq U_2 \Xi_{t_1,t_2} \frac{t_2 - t_1 + 1}{C\sqrt{T}}, \tag{13c}$$

where Inequality (13a) is equivalent to the condition $C \geq 56 \frac{MU_2}{\rho^2 \sqrt{T}}$, Inequality (13b) holds by Inequality (9) and Inequality (13c) is true since $\Xi_{t_1,t_2} \leq 2(1 + \lambda_{t_1,t_2})$.

$\boxed{\text{Prove } ③ < \frac{H(t_2 - t_1 + 1)}{7}}$ If $\eta \leq \frac{C\sqrt{T}}{56mU_3}$ holds then also $③ < \frac{H(t_2 - t_1 + 1)}{7}$ holds. Indeed:

$$\frac{H(t_2 - t_1 + 1)}{7} > \frac{HM}{7\rho^2 m\eta} \tag{14a}$$

$$\geq U_3 2 \frac{4HM}{\rho^2} \frac{1}{C\sqrt{T}} \tag{14b}$$

$$\geq U_3 2(1 + \lambda_{t_1,t_2}) \frac{1}{C\sqrt{T}} \tag{14c}$$

$$\geq U_3 \Xi_{t_1,t_2} \frac{1}{C\sqrt{T}}, \tag{14d}$$

where Inequality (14a) hold by Lemma 11, Inequality (14b) holds if condition $\eta \leq \frac{C\sqrt{T}}{56mU_3}$ holds, and Inequality (14c) and Inequality (14d) follow the same reasoning as Inequality (13b) and Inequality (13c).

$\boxed{\text{Prove } ④ < \frac{H(t_2 - t_1 + 1)}{7}}$ If $\eta \leq \frac{1}{56mU_1 C\sqrt{T}}$ holds then also $④ < \frac{H(t_2 - t_1 + 1)}{7}$ holds. Indeed:

$$\frac{H(t_2 - t_1 + 1)}{7} > \frac{HM}{7\rho^2 m\eta}$$
$$\geq U_1 2 \frac{4HM}{\rho^2} C\sqrt{T} \tag{15}$$
$$\geq U_1 2(1 + \lambda_{t_1,t_2}) C\sqrt{T}$$
$$\geq U_1 \Xi_{t_1,t_2} C\sqrt{T},$$

where Inequality (15) holds when condition $\eta \leq \frac{1}{56mU_1 C\sqrt{T}}$ also holds, and the rest of the inequalities follow a similar reasoning to the one used to bound $③$.

$\boxed{\text{Prove } ⑤ \leq \frac{H(t_2 - t_1 + 1)}{7}}$ It is immediate to see that if $\eta \leq \frac{H}{7D_2}$ holds, then it holds also that:

$$⑤ = D_2 \eta(t_2 - t_1 + 1) \leq \frac{H(t_2 - t_1 + 1)}{7}.$$

$\boxed{\text{Prove } ⑥ < \frac{H(t_2 - t_1 + 1)}{7}}$

If the condition $M \geq 112 D_1 Hm$ is satisfied than the inequality $⑥ < \frac{H(t_2 - t_1 + 1)}{7}$ holds too. Indeed:

$$\frac{H(t_2 - t_1 + 1)}{7} > \frac{HM}{7\rho^2 m\eta} \tag{16a}$$

$$\geq D_1 \frac{16H^2}{\rho^2} \frac{1}{\eta} \tag{16b}$$

$$\geq D_1 \frac{\|\lambda_{t_1}\|_1^2}{\eta} \tag{16c}$$

$$\geq D_1 \frac{\|\lambda_{t_1}\|_2^2}{\eta},$$

where Inequality 16a holds by Lemma 11, Inequality (16b) holds when the condition $M \geq 112 D_1 Hm$ is satisfied and Inequality (16c) holds by Inequality (8).

$\boxed{\text{Prove } \textcircled{7} < \frac{H(t_2 - t_1 + 1)}{7}}$

If the condition $\eta \leq \frac{1}{56mU_4\sqrt{T}}$ is satisfied then $\textcircled{7} < \frac{H(t_2 - t_1 + 1)}{7}$ also holds. In fact

$$\frac{H(t_2 - t_1 + 1)}{7} > \frac{HM}{7\rho^2 m\eta} \tag{17a}$$

$$\geq U_4 2\frac{4HM}{\rho^2}\sqrt{T} \tag{17b}$$

$$\geq U_4 2(1 + \lambda_{t_1,t_2})\sqrt{T}$$

$$\geq U_4 \Xi_{t_1,t_2}\sqrt{T}.$$

where Inequality (17a) holds by Lemma 11 and inequality (17b) holds if condition $\eta \leq \frac{1}{56mU_4\sqrt{T}}$ also holds.

**Conclusion of the proof** Thus, we have the following 3 conditions:

- First condition:
$$M \geq 112 D_1 Hm$$
$$= 112\frac{1}{2}Hm$$
$$= 56Hm.$$

- Second condition:
$$C \geq 56\frac{MU_2}{\rho^2\sqrt{T}}$$
$$= 56\frac{M9H|X||A|}{\rho^2\sqrt{T}}.$$

- Third condition:
$$\eta \leq \min\left\{\frac{1}{14m(F_1 + B_1)\sqrt{T}}, \frac{C\sqrt{T}}{56mU_3}, \frac{1}{56mU_1C\sqrt{T}}, \frac{H}{7D_2}, \frac{1}{56mU_4\sqrt{T}}\right\}.$$

Thus, we set $M$ as $M = 56Hm$, and consequently, under Condition 2 we set $C = 252|X||A|H$ since

$$C = 252|X||A|H$$
$$\geq 252|X||A|$$
$$\geq 252|X||A|\frac{112mH^2}{\rho^2}\frac{1}{\sqrt{T}}$$
$$= 56\frac{(56Hm)9H|X||A|}{\rho^2\sqrt{T}}$$
$$= 56\frac{9MH|X||A|}{\rho^2\sqrt{T}}.$$

Notice that the inequality is deduced directly by Condition 2. In fact if $\rho \geq T^{-\frac{1}{8}}H\sqrt{112m}$ then it is also true that

$$\frac{112mH^2}{\rho^2} \leq T^{\frac{1}{4}} \leq \sqrt{T}.$$

As a final remark, we choose $252|X||A|H$ as value of $C$ instead of the smaller value $252|X||A|$, which is useful for Lemma 6. Finally we study the condition on $\eta$.

$$\min\left\{\frac{1}{14m(F_1 + B_1)\sqrt{T}}, \frac{C\sqrt{T}}{56mU_3}, \frac{1}{56mU_1C\sqrt{T}}, \frac{H}{7D_2}, \frac{1}{56mU_4\sqrt{T}}\right\}$$

$$\geq \min \left\{ \frac{1}{14m\left(4H\sqrt{\ln\left(\frac{T^2}{\delta}\right)}\right)\sqrt{T}}, \frac{252|X||A|H\sqrt{T}}{56m\left(\frac{H}{2}\ln\left(\frac{HT^2}{\delta}\right)\right)}, \right.$$

$$\frac{1}{56m\left(6H^2\ln\left(\frac{H|A|T^2}{\delta}\right)\right)(252|X||A|H)\sqrt{T}}, \frac{H}{7\left(\frac{mH^2}{2}\right)},$$

$$\left. \frac{1}{56m\left(30H^2|X|^2\sqrt{2|A|\ln\left(\frac{T|X|^2|A|}{\delta}\right)}\right)\sqrt{T}} \right\}$$

$$\geq \frac{1}{84672mH^2|X|^2|A|\ln\left(\frac{|A||X|^2T^2}{\delta}\right)\sqrt{T}}.$$

Thus, the proof is concluded taking $\eta = \frac{1}{84672mH^2|X|^2|A|\ln\left(\frac{|A||X|^2T^2}{\delta}\right)\sqrt{T}}$. $\qquad\square$

**Lemma 3.** *If Condition 2 holds, for all $t \in [T]$ and for each constraints $i \in [m]$, it holds:*

$$\lambda_{t,i} \geq \eta\widehat{V}_{t-1,i},$$

*where $\widehat{V}_{t,i} := \sum_{\tau=1}^t \sum_{x,a} g_{\tau,i}(x,a)\mathbb{I}_\tau(x,a)$.*

*Proof.* First observe that with $t = 1$ we have that $\widehat{V}_{t-1,i}$ is the sum of zero elements and as such, it is equal to zero. This means that for $t = 1$ the inequality $\lambda_{t,i} \geq \eta\widehat{V}_{t-1,i}$ is equivalent to

$$\lambda_{t,i} \geq 0,$$

which is true by construction. We finish the proof by induction. Suppose $\lambda_{t,i} \geq \eta\widehat{V}_{t-1,i}$ is true for a $t \in [T]$, we show that it also holds for $t + 1$, indeed:

$$\lambda_{t+1,i} = \max\left\{\lambda_{t,i} + \eta\sum_{h=0}^{H-1} g_{t,i}(x_h,a_h), 0\right\}$$

$$= \max\left\{\lambda_{t,i} + \eta\sum_{x,a} g_{t,i}(x,a)\mathbb{I}_t(x,a), 0\right\}$$

$$\geq \lambda_{t,i} + \eta\sum_{x,a} g_{t,i}(x,a)\mathbb{I}_t(x,a)$$

$$\geq \eta\widehat{V}_{t-1,i} + \eta\sum_{x,a} g_{t,i}(x,a)\mathbb{I}_t(x,a)$$

$$= \eta\left(\sum_{\tau=1}^{t-1} g_{\tau,i}(x,a)\mathbb{I}_\tau(x,a) + g_{t,i}(x,a)\mathbb{I}_t(x,a)\right)$$

$$= \eta\sum_{\tau=1}^t g_{\tau,i}(x,a)\mathbb{I}_t(x,a)$$

$$= \eta\widehat{V}_{t,i}.$$

This concludes the proof. $\qquad\square$

**Lemma 4.** *If Condition 2 holds, referring as $i^*$ to the element in $[m]$ such that $i^* = \arg\max_{i\in[m]} \sum_{t=1}^T \left[G_t^\top q^{P,\pi_t}\right]_i$, then with probability at least $1 - \delta$, it holds:*

$$V_T \leq \widehat{V}_{T,i^*} + \mathcal{E}^{\mathbb{I}}.$$

*Proof.* We observe with probability at least $1 - \delta$:

$$V_T = \sum_{t=1}^T \left[G_t^\top q^{P,\pi_t}\right]_{i^*}$$

$$= \sum_{t=1}^{T} \sum_{x,a} g_{t,i^*}(x,a) \left( q_t(x,a) - \mathbb{I}_t(x,a) \right) + \sum_{t=1}^{T} \sum_{x,a} g_{t,i^*}(x,a) \mathbb{I}_t(x,a)$$

$$\leq \sum_{t=1}^{T} \sum_{x,a} g_{t,i^*}(x,a) \left( q_t(x,a) - \mathbb{I}_t(x,a) \right) + \widehat{V}_{T,i^*}$$

$$\leq \mathcal{E}^{\mathbb{I}} + V_{T,i^*}.$$

This concludes the proof. $\qquad\square$

**Lemma 5.** *When Condition 2 does not hold, with probability at least $1 - 10\delta$ in case of stochastic costs and $1 - 9\delta$ in case of adversarial costs it holds for all $i \in [m]$:*

$$\widehat{V}_{T,i} \leq \frac{4T^{\frac{1}{4}}}{\eta}.$$

*Proof.* Recall the definition of $\widehat{V}_{t,i}$ as $\widehat{V}_{t,i} = \sum_{\tau=1}^{t} \sum_{x,a} g_{t,i}(x,a) \mathbb{I}_t(x,a)$. We first focus on the stochastic setting. Thus, with probability at least $1 - \delta$, it holds:

$$\sum_{t=1}^{T} r_t^\top q_t - \sum_{t=1}^{T} \lambda_t^\top G_t^\top q_t = \sum_{t=1}^{T} r_t^\top q^\circ - \sum_{t=1}^{T} \lambda_t^\top G_t^\top q^\circ + \sum_{t=1}^{T} \ell_t^{\mathcal{L},\top}(q^\circ - q_t)$$

$$\geq -\sum_{t=1}^{T} \lambda_t^\top \overline{G}^\top q^\circ - \lambda_{1,T} \mathcal{E}_T^G + \sum_{t=1}^{T} \ell_t^{\mathcal{L},\top}(q^\circ - q_t)$$

$$\geq -mT^{\frac{1}{4}} \mathcal{E}_T^G + \sum_{t=1}^{T} \ell_t^{\mathcal{L},\top}(q^\circ - q_t).$$

On the other hand, in case of adversarial constraints, it holds:

$$\sum_{t=1}^{T} r_t^\top q_t - \sum_{t=1}^{T} \lambda_t^\top G_t^\top q_t = \sum_{t=1}^{T} r_t^\top q^\circ - \sum_{t=1}^{T} \lambda_t^\top G_t^\top q^\circ + \sum_{t=1}^{T} \ell_t^{\mathcal{L}\top}(q^\circ - q_t)$$

$$\geq \sum_{t=1}^{T} \ell_t^{\mathcal{L},\top}(q^\circ - q_t).$$

Define a vector $\widetilde{\lambda} \in [0, T^{\frac{1}{4}}]^m$ as $\widetilde{\lambda}_j = 0$ if $j \neq i$ and $\widetilde{\lambda}_j = T^{\frac{1}{4}}$ if $j = i$. Simultaneously with probability at least $1 - \delta$ it holds:

$$\sum_{t=1}^{T} r_t^\top q_t - \sum_{t=1}^{T} \lambda_t^\top G_t^\top q_t$$

$$\leq \sum_{t=1}^{T} r_t^\top q_t - \sum_{t=1}^{T} \widetilde{\lambda}^\top \sum_{x,a} G_t(x,a) \mathbb{I}_t(x,a) + \sum_{t=1}^{T} (\widetilde{\lambda} - \lambda_t)^\top \sum_{x,a} G_t(x,a) \mathbb{I}_t(x,a) + \lambda_{1,T} \mathcal{E}^{\mathbb{I}}$$

$$\leq \sum_{t=1}^{T} r_t^\top q_t - \sum_{t=1}^{T} \widetilde{\lambda}^\top \sum_{x,a} G_t(x,a) \mathbb{I}_t(x,a) + \mathcal{E}_T^D(\widetilde{\lambda}) + mT^{\frac{1}{4}} \mathcal{E}^{\mathbb{I}}$$

$$\leq HT - T^{\frac{1}{4}} \widehat{V}_{T,i} + \mathcal{E}_T^D(\widetilde{\lambda}) + mT^{\frac{1}{4}} \mathcal{E}^{\mathbb{I}},$$

where in the first equality we used the definition of $\mathcal{E}_T^{\mathbb{I}}$, in the first inequality we used the definition of the dual space $[0, T^{\frac{1}{4}}]^m$ to bound $\lambda_{1,T}$ as $mT^{\frac{1}{4}}$, and in the last inequality we used the definition of $\widetilde{\lambda}$. We can then compare the lower and the upper bound for $\sum_{t=1}^{T} r_t^\top q_t - \sum_{t=1}^{T} \lambda_t^\top G_t^\top q_t$ obtaining the following inequality, which holds with probability at least $1 - \delta$ with adversarial constraints and with probability at least $1 - 2\delta$ with stochastic constraints:

$$-mT^{\frac{1}{4}} \mathcal{E}_T^G + \sum_{t=1}^{T} \ell_t^{\mathcal{L},\top}(q^\circ - q_t) \leq HT - T^{\frac{1}{4}} \widehat{V}_{T,i} + \mathcal{E}_T^D(\widetilde{\lambda}) + mT^{\frac{1}{4}} \mathcal{E}^{\mathbb{I}},$$

from which we can write the following inequality that holds with probability at least $1 - 9\delta$ with adversarial constraints and $1 - 10\delta$ with stochastic constraints:

$$T^{\frac{1}{4}}\widehat{V}_{T,i} \leq mT^{\frac{1}{4}}(\mathcal{E}_T^G + \mathcal{E}_T^{\mathbb{I}}) + \mathcal{E}_T^P + \mathcal{E}_T^D(\widetilde{\lambda}) + HT. \tag{18}$$

We proceed now to bound each element of the right side of the inequality.

To bound $\mathcal{E}^P$ we use the fact that $\Xi_{1,T} \leq (1 + \lambda_{1,T}) \leq (1 + mT^{\frac{1}{4}})$ and the definition of $\eta$ as following:

$$\mathcal{E}_T^P = \Xi_{1,T}\left(U_1 C\sqrt{T} + U_2\frac{\sqrt{T}}{C} + U_3\frac{1}{C\sqrt{T}}\right) + U_4\sqrt{T}$$

$$\leq 2(1 + mT^{\frac{1}{4}})\sqrt{T}\left(1512H^3|X||A|\ln\left(\frac{H|A|T^2}{\delta}\right) + \frac{9H|X||A|}{252H|X||A|} + \frac{\frac{H}{2}\ln\left(\frac{HT^2}{\delta}\right)}{252H|X||A|}\right)$$

$$+ \sqrt{T}30H^2|X|^2\sqrt{2|A|\ln\left(\frac{T|X|^2|A|}{\delta}\right)}$$

$$\leq T^{\frac{1}{4}}\sqrt{T}6056H^3m|X||A|\ln\left(\frac{H|A|T^2}{\delta}\right) + \sqrt{T}30H^2|X|^2\sqrt{2|A|\ln\left(\frac{T|X|^2|A|}{\delta}\right)}$$

$$\leq T^{\frac{1}{4}}\sqrt{T}6116H^2m|X|^2|A|\ln\left(\frac{|X|^2|A|T^2}{\delta}\right)$$

$$\leq \frac{T^{\frac{1}{4}}}{\eta}$$

To bound $\mathcal{E}_T^D(\widetilde{\lambda})$ we use Theorem 11, the fact that by its definition $\|\widetilde{\lambda}\|_2^2 = \left(T^{\frac{1}{4}}\right)^2 = \sqrt{T}$, the initialization of the dual $\lambda_1 = \underline{0}$ and the definition of $\eta$ in the following way:

$$\mathcal{E}_T^D(\widetilde{\lambda}) \leq \frac{\|\lambda_1 - \widetilde{\lambda}\|_2^2}{2\eta} + \frac{\eta}{2}TmH^2$$

$$= \frac{\|\widetilde{\lambda}\|_2^2}{2\eta} + \frac{\eta}{2}TmH^2$$

$$= \frac{\sqrt{T}}{2\eta} + \frac{\eta}{2}TmH^2$$

$$\leq \frac{\sqrt{T}}{2\eta} + \frac{mH^2T}{2}\frac{1}{84672mH^2|X|^2|A|\ln\left(\frac{|A||X|^2T^2}{\delta}\right)\sqrt{T}}$$

$$\leq \frac{\sqrt{T}}{2\eta} + \frac{\sqrt{T}}{2\eta} = \frac{\sqrt{T}}{\eta}$$

We proceed to simply bound also $mT^{\frac{1}{4}}\left(\mathcal{E}_T^G + \mathcal{E}_T^{\mathbb{I}}\right)$ through their definition:

$$mT^{\frac{1}{4}}\left(\mathcal{E}_T^G + \mathcal{E}_T^{\mathbb{I}}\right) = mT^{\frac{1}{4}}\sqrt{T}\left(2H\sqrt{\ln\left(\frac{T^2}{\delta}\right)} + H\sqrt{2\ln\left(\frac{T^2}{\delta}\right)}\right)$$

$$\leq \frac{T^{\frac{1}{4}}}{\eta}$$

Finally we bound $HT$ as $HT \leq \frac{\sqrt{T}}{\eta}$.

Thus, Inequality (18) becomes

$$\widehat{V}_{T,i} \leq \frac{1}{T^{\frac{1}{4}}}\left(mT^{\frac{1}{4}}(\mathcal{E}_T^G + \mathcal{E}_T^{\mathbb{I}}) + \mathcal{E}_T^P + \mathcal{E}_T^D(\widetilde{\lambda}) + HT\right) \leq \frac{4\sqrt{T}}{T^{\frac{1}{4}}\eta} = \frac{4T^{\frac{1}{4}}}{\eta},$$

which concludes the proof. $\qquad\square$

## G   ANALYSIS WITH STOCHASTIC CONSTRAINTS

**Theorem 5.** *Suppose that Condition 2 holds and the constraints are generated stochastically. Then, for any $\delta \in (0, 1)$, Algorithm 2 attains:*

$$R_T \leq \widetilde{\mathcal{O}}\left(\Lambda\sqrt{T}\right), \quad V_T \leq \widetilde{\mathcal{O}}\left(\Lambda\sqrt{T}\right),$$

*with probability at least $1 - 14\delta$ when the rewards are stochastic and at least $1 - 13\delta$ when the rewards are adversarial.*

*Proof.* With probability at least $1 - 12\delta$ it holds:

$$V_T \leq \widehat{V}_{T,i^*} + \mathcal{E}^{\mathbb{I}} \tag{19a}$$

$$\leq \frac{1}{\eta}\lambda_{T+1,i^*} + \mathcal{E}^{\mathbb{I}} \tag{19b}$$

$$\leq \frac{1}{\eta}\Lambda + \mathcal{E}^{\mathbb{I}}, \tag{19c}$$

where Inequality (19a) holds by Lemma 4, Inequality (19b) holds by Lemma 3 and Inequality (19c) holds by Theorem 4. Then, with probability at least $1 - 12\delta$ we observe that:

$$\sum_{t=1}^{T} r_t^\top q^* - \sum_{t=1}^{T} r_t^\top q^{P,\pi_t} \leq \sum_{t=1}^{T}\left(r_t^\top q^* - \lambda_t^\top G_t^\top q^*\right) - \sum_{t=1}^{T}\left(r_t^\top q_t - \lambda_t^\top G_t^\top q_t\right) + \sum_{t=1}^{T}\lambda_t^\top G_t^\top\left(q^* - q_t\right)$$

$$\leq \mathcal{E}^P + \mathcal{E}^D(\underline{0}) + \lambda_{1,T}\mathcal{E}^{\mathbb{I}} + \sum_{t=1}^{T}\lambda_t^\top G_t^\top q^* \tag{20a}$$

$$= \mathcal{E}^P + \mathcal{E}^D(\underline{0}) + \lambda_{1,T}\mathcal{E}^{\mathbb{I}} + \sum_{t=1}^{T}\lambda_t^\top(G_t - \overline{G})^\top q^* + \sum_{t=1}^{T}\lambda_t^\top \overline{G}^\top q^*$$

$$\leq \mathcal{E}^P + \mathcal{E}^D(\underline{0}) + \lambda_{1,T}\mathcal{E}^{\mathbb{I}} + \lambda_{1,T}\mathcal{E}^G \tag{20b}$$

$$\leq \mathcal{E}^P + \mathcal{E}^D(\underline{0}) + \Lambda\mathcal{E}^{\mathbb{I}} + \Lambda\mathcal{E}^G, \tag{20c}$$

where Inequality (20a) holds by Theorem 3 and by Theorem 11, Inequality (20b) holds since in the stochastic constraint case $\sum_{t=1}^{T}(G_t - \overline{G})^\top q^* \leq \mathcal{E}^G$ with probability at least $1 - \delta$ by definition of $\mathcal{E}^G$, and finally Inequality (20c) holds by Theorem 4. Finally we observe that in the stochastic case with probability at least $1 - \delta$:

$$\left(T \cdot \text{OPT}_{\overline{r},\overline{G}} - \sum_{t=1}^{T} r_t^\top q_t\right) - \sum_{t=1}^{T} r_t^\top(q^* - q_t) \leq \mathcal{E}^r.$$

Thus, if the rewards are stochastic with probability at least $1 - 14\delta$ it holds:

$$R_T \leq \mathcal{E}^P + \mathcal{E}^D(\underline{0}) + \Lambda\mathcal{E}^{\mathbb{I}} + \Lambda\mathcal{E}^G + \mathcal{E}^r, \quad V_T \leq \frac{1}{\eta}\Lambda + \mathcal{E}^{\mathbb{I}}$$

and if the rewards are adversarial with probability at least $1 - 13\delta$ it holds:

$$R_T \leq \mathcal{E}^P + \mathcal{E}^D(\underline{0}) + \Lambda\mathcal{E}^{\mathbb{I}} + \Lambda\mathcal{E}^G, \quad V_T \leq \frac{1}{\eta}\Lambda + \mathcal{E}^{\mathbb{I}}$$

which concludes the proof. $\square$

**Theorem 6.** *Suppose that Condition 2 does not hold and the constraints are generated stochastically. Then, for any $\delta \in (0, 1)$, Algorithm 2 attains:*

$$R_T \leq \widetilde{\mathcal{O}}\left(T^{3/4}\right), \quad V_T \leq \widetilde{\mathcal{O}}\left(T^{3/4}\right),$$

*with probability at least $1 - 11\delta$ when the rewards are stochastic and at least $1 - 10\delta$ when the rewards are adversarial.*

*Proof.* Similar to the proof of Lemma 5 it holds with probability at least $1 - 10\delta$:

$$\sum_{t=1}^{T} r_t^\top q^* - \sum_{t=1}^{T} r_t^\top q^{P,\pi_t} \leq \sum_{t=1}^{T} \left( r_t^\top q^* - \lambda_t^\top G_t^\top q^* \right) - \sum_{t=1}^{T} \left( r_t^\top q_t - \lambda_t^\top G_t^\top q_t \right) + \sum_{t=1}^{T} \lambda_t^\top G_t^\top (q^* - q_t)$$

$$\leq \mathcal{E}^P + \mathcal{E}^D(\underline{0}) + \lambda_{1,T} \mathcal{E}^{\mathbb{I}} + \sum_{t=1}^{T} \lambda_t^\top G_t^\top q^*$$

$$= \mathcal{E}^P + \mathcal{E}^D(\underline{0}) + \lambda_{1,T} \mathcal{E}^{\mathbb{I}} + \sum_{t=1}^{T} \lambda_t^\top (G_t - \overline{G})^\top q^* + \sum_{t=1}^{T} \lambda_t^\top \overline{G}^\top q^*$$

$$\leq \mathcal{E}^P + \mathcal{E}^D(\underline{0}) + \lambda_{1,T} \mathcal{E}^{\mathbb{I}} + \lambda_{1,T} \mathcal{E}^G,$$

therefore with probability at least $1 - 10\delta$ following the reasoning of Lemma 5 it holds with adversarial rewards:

$$\sum_{t=1}^{T} r_t^\top q_t \geq T \cdot \mathrm{OPT}_{\overline{r}, \overline{G}} - m T^{1/4} \mathcal{E}^G + m T^{1/4} \mathcal{E}^{\mathbb{I}} - \mathcal{E}^D(\underline{0}) - \mathcal{E}^P,$$

and with stochastic rewards with probability ta least $1 - 11\delta$:

$$\sum_{t=1}^{T} r_t^\top q_t \geq T \cdot \mathrm{OPT}_{\overline{r}, \overline{G}} - m T^{1/4} \mathcal{E}^G + m T^{1/4} \mathcal{E}^{\mathbb{I}} - \mathcal{E}^D(\underline{0}) - \mathcal{E}^P - \mathcal{E}^r.$$

Applying Lemma 5 to bound the constraints violation concludes the proof. $\square$

## H ANALYSIS WITH ADVERSARIAL CONSTRAINTS

**Theorem 7.** *Suppose Condition 2 holds and the constraints are adversarial. Then, for any $\delta \in (0, 1)$, Algorithm 2 attains:*

$$\sum_{t=1}^{T} r_t^\top q_t \geq \Omega \left( \frac{\rho}{\rho + H} \cdot OPT_{\overline{r}, \overline{G}} \right), \quad V_T \leq \widetilde{\mathcal{O}} \left( \Lambda \sqrt{T} \right),$$

*with probability at least $1 - 14\delta$ when the rewards are stochastic and with probability at least $1 - 13\delta$ when the rewards are adversarial.*

*Proof.* Thanks to Theorem 3 , Theorem 11 and Theorem 4 with probability at least $1 - 11\delta$ it holds for all $q \in \Delta(P)$:

$$\sum_{t=1}^{T} r_t^\top q - \sum_{t=1}^{T} r_t^\top q_t$$

$$\leq -\sum_{t=1}^{T} \ell_t^{\mathcal{L},\top} q + \sum_{t=1}^{T} \ell_t^{\mathcal{L},\top} q_t + \sum_{t=1}^{T} \lambda_t^\top G_t^\top q - \sum_{t=1}^{T} \lambda_t^\top G_t^\top q_t$$

$$\leq \mathcal{E}_T^P + \sum_{t=1}^{T} \lambda_t^\top G_t^\top q + \sum_{t=1}^{T} (\underline{0} - \lambda_t)^\top \sum_{h=0}^{H-1} G_t(x_h, a_h) + \sum_{t=1}^{T} \lambda_t^\top \left( \sum_{h=0}^{H-1} G_t(x_h, a_h) - G_t^\top q_t \right)$$

$$\leq \mathcal{E}_T^P + \mathcal{E}_T^D(\underline{0}) + \lambda_{1,T} \mathcal{E}^{\mathbb{I}} + \sum_{t=1}^{T} \sum_{i=1}^{m} \lambda_{t,i} g_{t,i}^\top q$$

$$\leq \mathcal{E}_T^P + \mathcal{E}_T^D(\underline{0}) + \lambda_{1,T} \mathcal{E}^{\mathbb{I}} + \sum_{t=1}^{T} \sum_{i=1}^{m} \lambda_{t,i} g_{t,i}^\top q$$

$$\leq \mathcal{E}_T^P + \mathcal{E}_T^D(\underline{0}) + \Lambda \mathcal{E}^{\mathbb{I}} + \sum_{t=1}^{T} \sum_{i=1}^{m} \lambda_{t,i} g_{t,i}^\top q.$$

Consider now the occupancy measure $\widetilde{q} = \frac{\rho}{H+\rho}q^* + \frac{H}{H+\rho}q^\circ$. For all $i \in [m]$ and for all $t \in [T]$:

$$g_{t,i}^\top \widetilde{q} \leq \left( \frac{\rho}{H+\rho}g_{t,i}^\top q^* + \frac{H}{H+\rho}g_{t,i}^\top q^\circ \right)$$

$$\leq \left( \frac{H\rho}{H+\rho} - \frac{H\rho}{H+\rho} \right)$$

$$= 0,$$

given that $g_{t,i}^\top q^* \leq \|q^*\|_1 \leq H$, and $g_{t,i}^\top q^\circ \leq -\rho$ by definition of $q^\circ$ and by definition of $\rho$.

$$\sum_{t=1}^T r_t^\top \widetilde{q} = \sum_{t=1}^T \left( \frac{\rho}{H+\rho}r_t^\top q^* + \frac{H}{H+\rho}r_t^\top q^\circ \right)$$

$$\geq \frac{\rho}{H+\rho}\sum_{t=1}^T r_t^\top q^*,$$

since $r_t^\top q^\circ \geq 0$. Notice also that with adversarial rewards $\sum_{t=1}^T r_t^\top q^* = T \cdot \text{OPT}_{\overline{r},\overline{G}}$, while with stochastic rewards with probability at least $1-\delta$ it holds $\sum_{t=1}^T r_t^\top q^* \geq T \cdot \text{OPT}_{\overline{r},\overline{G}} - \mathcal{E}^r$, by definition of $\mathcal{E}^r$ and $\text{OPT}_{\overline{r},\overline{G}}$ for stochastic rewards. By reordering the terms we get that with probability at least $1 - 11\delta$

$$\sum_{t=1}^T r_t^\top q_t \geq \frac{\rho}{H+\rho}\sum_{t=1}^T r_t^\top q^* - \mathcal{E}_T^P - \mathcal{E}_T^D(\underline{0}) - \Lambda\mathcal{E}^{\mathbb{I}},$$

we can proceed to bound the regret in both cases: adversarial rewards and stochastic rewards.

With probability at least $1 - 11\delta$ with adversarial rewards it holds:

$$R_T = \sum_{t=1}^T r_t^\top q^* - \sum_{t=1}^T r_t^\top q_t$$

$$\leq \sum_{t=1}^T r_t^\top q^* - \left( \frac{\rho}{H+\rho}\sum_{t=1}^T r_t^\top q^* - \mathcal{E}_T^P - \mathcal{E}_T^D(\underline{0}) - \Lambda\mathcal{E}^{\mathbb{I}} \right)$$

$$\leq \frac{H}{H+\rho}\sum_{t=1}^T r_t^\top q^* + \mathcal{E}_T^P + \mathcal{E}_T^D(\underline{0}) + \Lambda\mathcal{E}^{\mathbb{I}}$$

$$\leq \frac{H}{H+\rho}T \cdot \text{OPT}_{\overline{r},\overline{G}} + \mathcal{E}_T^P + \mathcal{E}_T^D(\underline{0}) + \Lambda\mathcal{E}^{\mathbb{I}}.$$

With stochastic rewards it holds with probability at least $1 - 11\delta$:

$$\sum_{t=1}^T r_t^\top q_t \geq \frac{\rho}{H+\rho}\sum_{t=1}^T r_t^\top q^* - \mathcal{E}_T^P - \mathcal{E}_T^D(\underline{0}) - \Lambda\mathcal{E}^{\mathbb{I}},$$

and with probability at least $1 - 12\delta$:

$$\sum_{t=1}^T r_t^\top q_t \geq \frac{\rho}{H+\rho}T \cdot \text{OPT}_{\overline{r},\overline{G}} - \mathcal{E}_T^P - \mathcal{E}_T^D(\underline{0}) - \Lambda\mathcal{E}^{\mathbb{I}} - \mathcal{E}^r.$$

To conclude the proof we observe that following the analogous reasoning to Theorem 5 in case of adversarial constraints it also holds with probability at least $1 - 12\delta$:

$$V_T \leq \frac{1}{\eta}\Lambda + \mathcal{E}^{\mathbb{I}}.$$

$$\square$$

# I ANALYSIS WITH RESPECT TO THE WEAKER BASELINE

In this section we will study the guarantees of Algorithm 2 when the regret is computed with respect to a policy $q^*$ that respect the constraints at each episode, *i.e.* $g_t^\top q^* \le 0$ for all $i \in [m]$, for all $t \in [T]$.

**Theorem 8.** *Suppose that Condition 2 holds and the constraints are generated adversarially. Then, for any $\delta \in (0,1)$, Algorithm 2 attains:*

$$R_T \le \widetilde{\mathcal{O}}\left(\Lambda\sqrt{T}\right), \quad V_T \le \widetilde{\mathcal{O}}\left(\Lambda\sqrt{T}\right),$$

*with probability at least $1 - 13\delta$ when the rewards are stochastic and at least $1 - 12\delta$ when the rewards are adversarial.*

*Proof.* The violation can be bounded as in Theorem 5. Also similarly to Theorem 5 it holds with probability $1 - 12\delta$

$$\sum_{t=1}^T r_t^\top q^* - \sum_{t=1}^T r_t^\top q_t \le -\sum_{t=1}^T \ell_t^{\mathcal{L},\top} q^* + \sum_{t=1}^T \ell_t^{\mathcal{L},\top} q_t + \sum_{t=1}^T \lambda_t^\top G_t^\top q^* - \sum_{t=1}^T \lambda_t^\top G_t^\top q_t$$

$$\le \mathcal{E}_T^P + \sum_{t=1}^T (\underline{0} - \lambda_t)^\top \sum_{h=0}^{H-1} G_t(x_h, a_h) + \sum_{t=1}^T \lambda_t^\top \left( \sum_{h=0}^{H-1} G_t(x_h, a_h) - G_t^\top q_t \right)$$

$$\le \mathcal{E}_T^P + \mathcal{E}_T^D(\underline{0}) + \lambda_{1,T} \mathcal{E}^{\mathbb{I}}$$

$$\le \mathcal{E}_T^P + \mathcal{E}_T^D(\underline{0}) + \Lambda \mathcal{E}^{\mathbb{I}}.$$

Finally with stochastic rewards with probability at least $1 - \delta$:

$$T \cdot \mathrm{OPT}_{\overline{r},\overline{G}} - \sum_{t=1}^T r_t^\top q^* \le \mathcal{E}^r.$$

Therefore, with adversarial rewards it holds with probability at least $1 - 12\delta$:

$$R_T \le \mathcal{E}^P + \mathcal{E}^D(\underline{0}) + \Lambda \mathcal{E}^{\mathbb{I}}, \quad V_T \le \frac{1}{\eta}\Lambda + \mathcal{E}^{\mathbb{I}},$$

and with stochastic rewards it holds with probability at least $1 - 13\delta$:

$$R_T \le \mathcal{E}^P + \mathcal{E}^D(\underline{0}) + \Lambda \mathcal{E}^{\mathbb{I}} + \mathcal{E}^r, \quad V_T \le \frac{1}{\eta}\Lambda + \mathcal{E}^{\mathbb{I}},$$

which concludes the proof. □

**Theorem 9.** *Suppose that Condition 2 does not hold and the constraints are generated adversarially. Then, for any $\delta \in (0,1)$, Algorithm 2 attains:*

$$R_T \le \widetilde{\mathcal{O}}\left(T^{3/4}\right), \quad V_T \le \widetilde{\mathcal{O}}\left(T^{3/4}\right),$$

*with probability at least $1 - 12\delta$ when the rewards are stochastic and at least $1 - 11\delta$ when the rewards are adversarial.*

*Proof.* The violation can be bounded thanks to Lemma 3, as in Theorem 6. To bound the regret, notice that it holds with probability $1 - 9\delta$:

$$\sum_{t=1}^T r_t^\top q^* - \sum_{t=1}^T r_t^\top q_t \le -\sum_{t=1}^T \ell_t^{\mathcal{L},\top} q^* + \sum_{t=1}^T \ell_t^{\mathcal{L},\top} q_t + \sum_{t=1}^T \lambda_t^\top G_t^\top q^* - \sum_{t=1}^T \lambda_t^\top G_t^\top q_t$$

$$\le \mathcal{E}_T^P + \sum_{t=1}^T (\underline{0} - \lambda_t)^\top \sum_{h=0}^{H-1} G_t(x_h, a_h) + \sum_{t=1}^T \lambda_t^\top \left( \sum_{h=0}^{H-1} G_t(x_h, a_h) - G_t^\top q_t \right)$$

$$\le \mathcal{E}_T^P + \mathcal{E}_T^D(\underline{0}) + \lambda_{1,T} \mathcal{E}^{\mathbb{I}}$$

$$\leq \mathcal{E}_T^P + \mathcal{E}_T^D(\underline{0}) + mT^{1/4}\mathcal{E}^{\mathbb{I}}.$$

Finally with stochastic rewards with probability at least $1 - \delta$:

$$T \cdot \text{OPT}_{\overline{r},\overline{G}} - \sum_{t=1}^{T} r_t^\top q^* \leq \mathcal{E}^r.$$

Therefore, with adversarial rewards it holds with probability at least $1 - 11\delta$:

$$R_T \leq \mathcal{E}^P + \mathcal{E}^D(\underline{0}) + mT^{1/4}\mathcal{E}^{\mathbb{I}}, \quad V_T \leq \frac{4T^{1/4}}{\eta},$$

and with stochastic rewards it holds with probability at least $1 - 12\delta$:

$$R_T \leq \mathcal{E}^P + \mathcal{E}^D(\underline{0}) + mT^{1/4}\mathcal{E}^{\mathbb{I}} + \mathcal{E}^r, \quad V_T \leq \frac{4T^{1/4}}{\eta},$$

which concludes the proof. $\qquad\square$

## J   AUXILIARY LEMMAS

**Lemma 6** (Adapted from (Luo et al., 2021) Lemma C.4).

$$\eta_t \widehat{Q}_t(x,a) \leq \frac{1}{2} \quad \wedge \quad \eta_t B_t(x,a) \leq \frac{1}{2H}.$$

*Proof.* Recall $\gamma = 2\eta_t H \Xi_t$. Thus, it holds:

$$\eta_t \widehat{Q}_t(x,a) \leq \frac{\eta_t H \Xi_t}{\gamma} = \frac{\eta_t H \Xi_t}{2\eta_t H \Xi_t} = \frac{1}{2}$$

and

$$\eta_t b_t(x,a) = \frac{3\eta_t H \Xi_t \gamma + \eta_t \Xi_t H(\overline{q}_t(x,a) - \underline{q}_t(x,a))}{\overline{q}_t(x,a) + \gamma} \leq 3\eta_t \Xi_t H + \eta_t H \Xi_t = 2\gamma.$$

Finally,

$$\begin{aligned}
\eta_t B_t(x,a) &\leq H\left(1 + \frac{1}{H}\right)^H \eta_t \sup_{x',a'} b_t(x',a') \\
&\leq 3H2\gamma \\
&= 6H\gamma \\
&= \frac{6H}{C\sqrt{T}} \\
&= \frac{6H}{252|X||A|H\sqrt{T}} \\
&\leq \frac{1}{42H} \\
&\leq \frac{1}{2H}.
\end{aligned}$$

This concludes the poof. $\qquad\square$

**Lemma 7** (Adapted from (Luo et al., 2021), Lemma B.1). *If the following inequality holds:*

$$\sum_x q^*(x) \sum_{t=t_1}^{t_2} \sum_a \left(\pi_t(a|x) - \pi^*(a|x)\right)\left(Q_t^{\pi_t}(x,a) - B_t(x,a)\right)$$

$$\leq o(T) + \sum_{t=t_1}^{t_2} V^{\pi^*}(x_0; b_t) + \frac{1}{H} \sum_{t=t_1}^{t_2} \sum_{x,a} q^*(x)\pi_t(a|x)B_t(x,a), \tag{21}$$

with $B_t$ defined as

$$B_t(x,a) = b_t(x,a) + \left(1 + \frac{1}{H}\right) \mathbb{E}_{x' \sim P(\cdot|x,a)} \mathbb{E}_{a' \sim \pi_t(\cdot|x')} \left[B_t(x',a')\right] \quad \forall t \in [T], \forall x \in X, \forall a \in A,$$
(22)

then it holds that:

$$R_{t_1,t_2} \leq o(T) + 3 \sum_{t=t_1}^{t_2} \widehat{V}^{\pi_t}(x_0; b_t).$$

*Proof.* The proof is analogous to the one proposed by (Luo et al., 2021), Lemma B.1, since the proof is episode based and then the sum over $t$ is taken. □

**Lemma 8** (Adapted from (Luo et al., 2021), Lemma A.1). *Let $\mathcal{F}_0, \ldots, \mathcal{F}_T$ be a filtration and $X_1, \ldots, X_T$ be real random variables such that $X_t$ is $\mathcal{F}_t$-measurable, $\mathbb{E}[X_t|\mathcal{F}_t] = 0$, $|X_t| \leq b$ for all $t \in [T]$ and $\sum_{t=t_1}^{t_2} \mathbb{E}[X_t^2|\mathcal{F}_t] \leq V_{t_1,t_2}$ for some fixed $V_{t_1,t_2} > 0$ and $b > 0$ for every $t_1, t_2 \in [T]$ such that $1 \leq t_1 \leq t_2 \leq T$. Then with probability at least $1 - \delta$ it holds simultaneously for all $[t_1, \ldots, t_2] \subset [T]$:*

$$\sum_{t=t_1}^{t_2} X_t \leq \frac{V_{t_1,t_2}}{b} + b \log \left(\frac{T^2}{\delta}\right).$$

*Proof.* For all $\delta' \in (0,1)$ by Lemma A.1 (Luo et al., 2021) it holds:

$$\mathbb{P}\left(\sum_{t=t_1}^{t_2} X_t \geq \frac{V_{t_1,t_2}}{b} + b \log \left(\frac{1}{\delta'}\right)\right) \leq \delta'.$$

It is sufficient to consider the intersection of all events for all possible intervals $[t_1, \ldots, t_2]$, that are less than $T^2$.

$$\mathbb{P}\left(\bigcap_{t_1,t_2} \left\{\sum_{t=t_1}^{t_2} X_t \geq \frac{V_{t_1,t_2}}{b} + b \log \left(\frac{1}{\delta'}\right)\right\}\right) \leq T^2 \delta'.$$

To conclude the proof we take $\delta$ as $T^2 \delta'$. □

Consider a loss function $f_t(x,a) \in [0, Z]$, for all $t \in [T], (x,a) \in X \times A$, with $Z > 0$. Define another function $\widetilde{f}_t \in [0, Z]^{|X \times A|}$. If we define the estimator $\widehat{f}_t(x,a) = \frac{\widetilde{f}_t(x,a)\mathbb{I}_t(x,a)}{\overline{q}_t(x,a)+\gamma}$ where $\mathbb{E}[\widehat{f}_t(x,a)] = f_t(x,a)$, we can state the following result.

**Lemma 9** (Adapted from (Jin et al., 2020)). *For every sequence of functions $\alpha_1, \ldots \alpha_T$ such that $\alpha_t \in [0, \frac{2\gamma}{Z}]^{|X \times A|}$ is $\mathcal{F}_t$ measurable for all $t \in [T]$, we have with probability at least $1 - \delta$ that simultaneously for all $t_1, t_2 \in [T]$ such that $1 \leq t_1 \leq t_2 \leq T$ it holds:*

$$\sum_{t=t_1}^{t_2} \sum_{x,a} \alpha_t(x,a) \left(\widehat{f}_t(x,a) - \frac{q_t(x,a)}{\overline{q}_t(x,a)} f_t(x,a)\right) \leq H \ln \left(\frac{HT^2}{\delta}\right).$$

*Proof.*

$$\widehat{\ell}_t(x,a) = \frac{\widetilde{f}_t(x,a)\mathbb{I}_t(x,a)}{\overline{q}_t(x,a) + \gamma}$$

$$\leq \frac{\widetilde{f}_t(x,a)\mathbb{I}_t(x,a)}{\overline{q}_t(x,a) + \frac{\widetilde{f}_t(x,a)}{Z}\gamma}$$

$$= \frac{\mathbb{I}_t(x,a)Z}{2\gamma} \frac{2\gamma \frac{\widetilde{f}_t(x,a)}{Z}}{\overline{q}_t(x,a) + \gamma\frac{\widetilde{f}_t(x,a)}{Z}}$$

$$= \frac{\mathbb{I}_t(x,a)Z}{2\gamma} \frac{2\gamma \frac{\widetilde{f}_t(x,a)}{Z\overline{q}_t(x,a)}}{1 + \gamma\frac{\widetilde{f}_t(x,a)}{Z\overline{q}_t(x,a)}}$$

$$\leq \frac{Z}{2\gamma} \ln \left( 1 + 2\gamma \frac{\mathbb{I}_t(x,a)\widetilde{f}_t(x,a)}{Z\overline{q}_t(x,a)} \right).$$

For each layer $h \in [H]$ we define $\widehat{S}_{t,h} := \sum_{x \in X_h, a \in A} \alpha_t(x,a)\widehat{f}_t(x,a)$ and $S_{t,h} := \sum_{x \in X_h, a \in A} \alpha_t(x,a)\frac{q_t(x,a)}{\overline{q}_t(x,a)}f_t(x,a)$.

$$\mathbb{E}_t[\exp(\widehat{S}_{t,h})] = \mathbb{E}\left[ \exp\left( \sum_{x \in X_h, a \in A} \alpha_t(x,a)\widehat{f}_t(x,a) \right) \right]$$

$$\leq \mathbb{E}\left[ \exp\left( \sum_{x \in X_h, a \in A} \alpha_t(x,a)\frac{Z}{2\gamma}\ln\left(1 + 2\gamma\frac{\mathbb{I}_t(x,a)\widetilde{f}_t(x,a)}{Z\overline{q}_t(x,a)}\right) \right) \right]$$

$$\leq \mathbb{E}\left[ \prod_{x \in X_h, a \in A} \left( 1 + \alpha_t(x,a)\frac{\mathbb{I}_t(x,a)\widetilde{f}_t(x,a)}{\overline{q}_t(x,a)} \right) \right]$$

$$\leq 1 + \sum_{x \in X_h, a \in A} \alpha_t(x,a)\frac{q_t(x,a)f_t(x,a)}{\overline{q}_t(x,a)} = 1 + S_{t,h}$$

$$\leq \exp(S_{t,h})$$

For each interval $[t_1, \dots, t_2] \subset [T]$ it holds:

$$\mathbb{P}\left[ \sum_{t=t_1}^{t_2}(\widehat{S}_{t,h} - S_{t,h}) \geq \ln\left(\frac{H}{\delta'}\right) \right] \leq \frac{\delta'}{H}.$$

Taking the intersection event for all intervals $[t_1, \dots, t_2] \subset [T]$:

$$\mathbb{P}\left[ \bigcap_{t_1,t_2} \left\{ \sum_{t=t_1}^{t_2}(\widehat{S}_{t,h} - S_{t,h}) \geq \ln\left(\frac{H}{\delta'}\right) \right\} \right] \leq T^2\frac{\delta'}{H}.$$

$$\delta = T^2\delta',$$

and

$$\mathbb{P}\left[ \bigcap_{t_1,t_2} \left\{ \sum_{t=t_1}^{t_2}(\widehat{S}_{t,h} - S_{t,h}) \geq \ln\left(\frac{HT^2}{\delta}\right) \right\} \right] \leq \frac{\delta}{H}.$$

Finally we take the sum over $h \in [H]$:

$$\mathbb{P}\left[ \sum_{t=t_1}^{t_2}\sum_{x,a}\alpha_t(x,a)\left( \widehat{f}_t(x,a) - \frac{q_t(x,a)}{u_t(x,a)}f_t(x,a) \right) \leq H\ln\left(\frac{HT^2}{\delta}\right) \right] \leq \delta.$$

This concludes the proof. $\qquad\square$

**Corollary 1.** *Given $\delta \in (0,1)$, it holds with probability at least $1 - 2\delta$ simultaneously for all $t_1, t_2 \in [T]$ such that $1 \leq t_1 \leq t_2 \leq T$:*

$$\sum_{t=t_1}^{t_2}\sum_{x,a}\left( \widehat{f}_t(x,a) - f_t(x,a) \right) \leq \frac{ZH}{2\gamma}\ln\left(\frac{HT^2}{\delta}\right).$$

**Lemma 10.** *Let $\{\pi_t\}_{t=1}^T$ policies, then for any collection of transition $P_t^x \in \mathcal{P}_{i(t)}$ with probability at least $1 - 2\delta$,*

$$\sum_{t=1}^{T}\|q^{P,\pi_t} - q^{P_t^x,\pi_t}\|_1 \leq 2H|X|^2\sqrt{2T\ln\left(\frac{H|X|}{\delta}\right)} + 3H|X|^2\sqrt{2T|A|\ln\left(\frac{T|X|^2|A|}{\delta}\right)}.$$

*Proof.* It holds:

$$\sum_{t=1}^{T}\|q^{P,\pi_t} - q^{P_t^x,\pi_t}\|_1 = \sum_{t=1}^{T}\sum_{x,a}|q^{P,\pi_t}(x,a) - q^{P_t^x,\pi_t}(x,a)|$$

$$\leq \sum_{t=1}^{T}\sum_{x,a}\sum_{x'}|q^{P,\pi_t}(x',a) - q^{P_t^x,\pi_t}(x',a)|$$

$$= \sum_{x}\sum_{t=1}^{T}\sum_{x',a}|q^{P,\pi_t}(x',a) - q^{P_t^x,\pi_t}(x',a)|$$

$$\leq \sum_{x}\left(2H|X|\sqrt{2T\ln\left(\frac{H|X|}{\delta}\right)} + 3H|X|\sqrt{2T|A|\ln\left(\frac{T|X|^2|A|}{\delta}\right)}\right)$$

$$\leq |X|\left(2H|X|\sqrt{2T\ln\left(\frac{|X|H}{\delta}\right)} + 3H|X|\sqrt{2T|A|\ln\left(\frac{T|X|^2|A|}{\delta}\right)}\right),$$

by Lemma 13, taking the union bound over $X$ $\left(\delta' = \frac{\delta}{|X|}\right)$. This concludes the proof. $\square$

## K  AUXILIARY LEMMAS FROM EXISTING WORKS

**Lemma 11** (Stradi et al. (2024b) lemma D.2). *For $\eta \leq \frac{1}{mH}$ and $\frac{M}{\rho} > 4$, if $\|\lambda_{t_2+1}\|_1 > \frac{2HM}{\rho^2}$ and $\|\lambda_{t_1}\|_1 \leq \frac{4H}{\rho}$ it holds:*

$$(t_2 - t_1 + 1) > \frac{M}{\rho^2 m\eta}$$

**Lemma 12** (Rosenberg & Mansour (2019b)). *For any $\delta \in (0,1)$*

$$\|P(\cdot|x,a) - \overline{P}_i(\cdot|x,a)\|_1 \leq \sqrt{\frac{2|X_{h(x)+1}|\ln\left(\frac{T|X||A|}{\delta}\right)}{\max\{1, N_i(x,a)\}}},$$

*simultaneously for all $(x,a) \in X \times A$ and for all epochs with probability at least $1 - \delta$.*

**Lemma 13** (Rosenberg & Mansour (2019b)). *Let $\{\pi_t\}_{t=1}^{T}$ policies and let $\{P_t\}_{t=1}^{T}$ transition functions such that $q^{P_t,\pi_t} \in \Delta(\mathcal{P}_{i(t)})$ for every $t \in [T]$. Then with probability at least $1 - 2\delta$,*

$$\sum_{t=1}^{T}\|q^{P,\pi_t} - q^{P_t,\pi_t}\|_1 \leq 2H|X|\sqrt{2T\ln\left(\frac{H}{\delta}\right)} + 3H|X|\sqrt{2T|A|\ln\left(\frac{T|X||A|}{\delta}\right)}.$$

