# OpenReview forum: "Best-of-Both-Worlds Policy Optimization for CMDPs with Bandit Feedback"
_ICLR.cc/2025/Conference — Submitted to ICLR 2025_

### Official Review · Reviewer_NJCJ · 2024-11-03

**Soundness:** 2
**Presentation:** 2
**Contribution:** 2
**Rating:** 3
**Confidence:** 4

**Summary:**

This paper studied the constrained Markov decision process (CMDP) problem where the transition kernel is stochastic unknown, and the reward and costs could be adversarial and stochastic settings. The paper leverages the techniques from the existing literature to address the trade-off between rewards and constraint violations. The theoretical analysis is provided to justify the proposed algorithm.

**Strengths:**

The proposed algorithm achieves some interesting results, such as sublinear regret and constraint violation for stochastic settings and a constant competitive ratio for adversarial settings.

**Weaknesses:**

1. The paper studied the Constrained Markov Decision Process (CMDP) problem in both stochastic and adversarial settings. However, it heavily relies on the framework established by previous works (e.g., policy optimization in Luo et al., 2021; primal-dual design and analysis in Castiglioni et al., 2022b; Stradi et al., 2024b). This combination seems not interesting and the technical contribution of this paper is somewhat limited for a theory paper, especially considering the extensive literature on CMDPs, both in theory and algorithmic sides.

2. The metric of cumulative constraint violation is problematic in safety-critical applications in the paper because it may have limitations since it allows for compensation among different rounds.

3. The proposed method adopts a model-based approach and is limited to tabular settings. It might not be scalable to the setting with a large state and action space.

4. The paper primarily focuses on theoretical aspects; however, it would be beneficial to include numerical experiments to validate the proposed algorithm. By comparing it with representative studies in the extensive literature, such as the LP-based method discussed in Efroni et al. (2020) and Stradi et al. (2024b), as well as a model-based RL method in reference [1] and a model-free RL method in reference [2], the practical performance of the algorithm can be more clearly demonstrated. BTW, [1] and [2] are very related work and not mentioned in the paper.

[1] Tao Liu, Ruida Zhou, Dileep Kalathil, PR Kumar, and Chao Tian. Learning policies with zero or bounded constraint violation for constrained MDPs. NeurIPS 2021.

[2] Arnob Ghosh, Xingyu Zhou, and Ness Shroff. Provably efficient model-free constrained rl with linear function approximation. NeurIPS 2022.

**Questions:**

Please see the weakness.

---

### Official Review · Reviewer_S5u8 · 2024-11-03

**Soundness:** 3
**Presentation:** 2
**Contribution:** 2
**Rating:** 3
**Confidence:** 3

**Summary:**

This paper studies the problem of learning a constrained MDP under bandit feedback. The authors propose a policy-based algorithm to reach $\tilde{O}(T^{3/4})$ regret and violations. By assuming that the Lagrangian multiplier  is bounded by $\Lambda$, the regret and violation bounded could be improved to $\tilde{O}(\Lambda\sqrt{T})$.  In the meantime, the proposed algorithm could also achieve similar regret and violation bounds under the adversarial setting, which is referred as "best-of-both-worlds" property.

**Strengths:**

1. This paper considers the bandit feedback setting of learning a CMDP, which is the case of many practical problems..

2. The proposed regret\& violation bounds match the best previous result under full-information feedback.

**Weaknesses:**

1. The technical novelty is limited given the previous work (Stradi et. al., 2024). (Stradi et. all 2024) worked on the same problem with full-information feedback. As far as I can see, there is no substantial difficulty to combine their algorithm with classical estimators for online learning with bandit-feedback.

2. The key problem in (Stradi et. al., 2024) is not resolved (whether it is possible to remove the slater condition number in the violation and regret bound. In other words, whether a $\tilde{O}(\sqrt{T})$ regret & violation bound is achievable without further assumptions for the best-of-both-worlds setting). It is known (Jin et. al., 2020) that for reward-adv MDP without constraints, it is possible to reach an $\tilde(O)(\sqrt{T})$ regret, while for stochastic reward and constraints, there exists algorithm (Agrawal et. al., 2022) to reach $\tilde{O}(\sqrt{T})$ regret\& violation bound. So the question is that: is it possible to get an $\tilde{O}(\sqrt{T})$ regret (violation) bound for CMDP with adv reward (constraints)?

3. As claimed by the authors, one possible benefit by policy optimization is to avoid solving some convex optimization problem. But there is empirical results to demonstrate that the policy-based algorithm is more efficient than the occupancy-based algorithm.

References:

[Jin et. al., 2020] Learning Adversarial MDPs with Bandit Feedback and Unknown Transition

[Agrawal et. al., 2022] Regret Guarantees for Model-Based Reinforcement Learning with Long-Term Average Constraints

**Questions:**

I do not understand why the authors design  the algorithm based on policy optimization. It seems there is no substantial difficulty in applying the occupancy-based algorithm (Stradi et. al., 2024) with the same reward\&constraint estimators. Although the policy optimization algorithm seems more practical, it might leads to worse dependence on other parameters. Is there any potential advantages by applying policy optimization to solve CMDP?

Typo: it should be $\rho\geq T^{-1/8}H\sqrt{112m}$ in Condition 2?

---

> ### Comment · Reviewer_S5u8 · 2024-11-25
> **Discussion.**
>
> Thanks for the replay. I think the current  $\rho$ condition is correct. However, I am not convinced by the advantage of policy optimization so the technical novelty is still a major problem. As far as I know, the issues about occupancy based algorithm could be resolved by paying more regret.

---

> > ### Author Response · Authors · 2024-11-26
> >
> > We thank the Reviewer for the answer and the possibility of further clarifying the reason behind our choice of a policy optimization approach, as it is, as pointed out by the Reviewer, a crucial point for the novelty of the paper.
> >
> > To build an algorithm with the no-interval regret property, either policy based or occupancy measure based, we have to introduce a form of regularization to incentive the exploration. In our case we used the fixed-share update, which, when applied directly on a policy has an immediate interpretation: in each episode, given a state, each action should be played with at least a fixed threshold probability. Consider now the analogous on the occupancy measure space, the fixed share update would be of the form:
> > $\widetilde{q}\_t=(1-\sigma)\widehat{q}\_t+ \sigma \bar{q}$,
> > where $\widehat{q}\_t$ is the solution of the convex problem $\arg\min\_{q \in \Delta(\mathcal{P}\_i)}(D(\widetilde{q}\_{t-1}\|\|q)+\eta\_{t-1}\langle q,\widehat{\ell}\_{t-1}\rangle)$ ($\Delta(\mathcal{P}\_i)$ is the set of occupancy for which the transition probability associated is in the confidence interval of transition probabilities) and $\bar{q}(x,a)=\frac{1}{|X||A|}$ for all $x,a$. However, in general $\bar{q}$ is feasible only at first round, when the confidence interval of the transition probabilities cover the whole space of possible transition probabilities, while for a generic $t_1$ there are no guarantees that $\bar{q}$ is a feasible occupancy measure for episode $t_1$ and therefore there is no guarantee that the probability transition induced by $\widetilde{q}\_{t_1}$ belong to the confidence interval at episode $t_1$ of the transition probabilities. On the other hand, not using the fixed share update would lead to a Bregman divergence $D(q_{t_1}\|\|q_{t_2})$ possibly unbounded, which implies an impossibility of proving the no-interval regret property desired. As a side note, even when not considering the no-interval regret property, we believe that the policy optimization approach is to be preferred as it is computationally more efficient.

---

### Official Review · Reviewer_LZJr · 2024-11-04

**Soundness:** 2
**Presentation:** 3
**Contribution:** 2
**Rating:** 3
**Confidence:** 4

**Summary:**

This paper proposes an algorithm for online CMDPs with bandit feedback under both stochastic and adversarial constraints settings. Applying an existing policy optimization algorithm to a primal-dual approach to policy optimization, the algorithm achieves $\sqrt{T}$ regret and constraint violation bounds for stochastic constraints under certain conditions, and a $\sqrt{T}$ condition violation under adversarial constraints.

**Strengths:**

The paper considers the CMDP setting with bandit feedback in both stochastic and adversarial settings, which is more practical. The authors provide theoretical guarantees on regret and constraint violation, with a specific analysis of the dependency on Slater’s condition. By focusing on policy space optimization rather than occupancy measures, the paper presents a more computationally efficient alternative for real-world applications where solving convex programs per episode is infeasible.

**Weaknesses:**

- The paper’s theoretical results rely heavily on Condition 2. As discussed in Section 2.4, $\rho \in [0, H]$; however, it is unclear how this condition could hold even when $T = 1$. Even setting aside this issue, if Condition 2 does hold, it implies that $\rho$ must be a relatively large value as $T$ increases, which is likely to be the case in online settings. According to Theorem 4, the Lagrangian multiplier is bounded by $c/\rho^2$, resulting in a very small constant close to zero. In such cases, the reward function becomes the dominant term, yielding a regret rate of $\sqrt{T}$.

- The theoretical analysis heavily relies on prior work by Stradi et al. (2024b) and Luo et al. (2021).
- No simulations are included to support the efficiency claims made by the authors.

A lot of relevant papers on online CMDPs are missing:

[1] Arnob Ghosh, Xingyu Zhou, and Ness Shroff. Provably efficient model-free constrained rl with linear function approximation. Advances in Neural Information Processing Systems, 35:13303–13315, 2022

[2] Tao Liu, Ruida Zhou, Dileep Kalathil, Panganamala Kumar, and Chao Tian. Learning policies with zero or bounded constraint violation for constrained mdps. Advances in Neural Information Processing Systems, 34:17183–17193, 2021.

[3] Adrian Müller, Pragnya Alatur, Giorgia Ramponi, and Niao He. Cancellation-free regret bounds for lagrangian approaches in constrained markov decision processes. arXiv preprint arXiv:2306.07001, 2023.

[4] Adrian Müller, Pragnya Alatur, Volkan Cevher, Giorgia Ramponi, and Niao He. Truly no-regret learning in constrained mdps. arXiv preprint arXiv:2402.15776, 2024.

[5] Honghao Wei, Xin Liu, and Lei Ying. Triple-q: A model-free algorithm for constrained reinforcement learning with sublinear regret and zero constraint violation. In International Conference on Artificial Intelligence and Statistics, pages 3274–3307. PMLR, 2022.

[6] Qinbo Bai, Amrit Singh Bedi, Mridul Agarwal, Alec Koppel, and Vaneet Aggarwal. Achieving zero constraint violation for constrained reinforcement learning via primal-dual approach. In Proceedings of the AAAI Conference on Artificial Intelligence, volume 36, pages 3682–3689, 2022.

[7] Dongsheng Ding, Xiaohan Wei, Zhuoran Yang, Zhaoran Wang, and Mihailo Jovanovic. Provably efficient safe exploration via primal-dual policy optimization. In International conference on artificial intelligence and statistics, pages 3304–3312. PMLR, 2021.

[8] Dongsheng Ding, Chen-Yu Wei, Kaiqing Zhang, and Alejandro Ribeiro. Last-iterate convergent policy gradient primal-dual methods for constrained mdps. Advances in Neural Information Processing Systems, 36, 2024.

**Questions:**

If Condition 2 holds, then the regret and violation are $\Lambda \sqrt{T}$. According to Theorem 4, $\Lambda \leq \frac{c}{\rho^2} \leq \frac{c}{T^4}$. Does this mean that the regret and violation can achieve an order of $T^{1/4}$? How can this be true, as it would even better than the lower bound?

---

### Official Review · Reviewer_6xb1 · 2024-11-04

**Soundness:** 3
**Presentation:** 4
**Contribution:** 3
**Rating:** 6
**Confidence:** 2

**Summary:**

The authors introduce an algorithm for online learning in CMDP with either stochastic and adversarial constraints while only requiring bandit feedback unlike in prior work. The resulting regret and constraint violation bounds matches prior work with full-feedback and retain the property of not requiring the knowledge of the Slater's parameter. Additionally, the authors show that using a weaker-baseline that only requires a policy to satisfy a constraint at each episode, the propose algorithm achieve sub-linear regret and constraint violations.

**Strengths:**

- The proposed method extends prior work to a more practical setting while retaining similar guarantees
- The authors do a good job in presenting the problem setting and algorithm

**Weaknesses:**

- It's unclear why is the useful to consider a weaker baseline since it is unclear what is the befit (from a practical standpoint) of only considering the optimal policy that  respect the constraints at each episode
- Having simple numerical experiments would help showcase the benefit of updating the primal based on a policy optimization approach vs the occupancy measure space.

**Questions:**

- Besides allowing the algorithm to be more efficient by, is there any theoretical benefit of conducting the analysis via an policy optimization approach. For example, could the analysis have been done using an occupancy-measure-based methods like in prior work?
- Spelling: 176 "ti" -> "it"

---

### Official Review · Reviewer_CXZu · 2024-11-04

**Soundness:** 3
**Presentation:** 3
**Contribution:** 2
**Rating:** 3
**Confidence:** 4

**Summary:**

The authors present an algorithm for adversarial CMDP. They showed that it achieves $\mathcal{O}(T^{3/4})$ regret and constraint violation both in the stochastic and adversarial setting with modified baselines.

**Strengths:**

1. Clarity on presentation.
2. Sound analysis.

**Weaknesses:**

**Minor Comments**

1. I think the definition of the occupancy measure is wrong. See the questions below for more details.

2. The bonus update seems ambiguous (lines 312, 313). In this update, both LHS and RHS have the term $B_t(x, a)$, and the only initial condition given is $B_t(x_H, a)=0$, $\forall a$ (this is mentioned in the algorithm, not in the main text). I guess the layered structure of the state space is essential to fill up all entries of $B_t(\cdot, \cdot)$ using the stated update. This should be emphasized in the main text for the reader's benefit.

**Major Comments**

3. Theorem 3: In the worst case, the loss range upper bound can be an increasing function of $T$. Hence, in general, the regret upper bound may no longer be $\mathcal{O}(\sqrt{T})$, and thus the algorithm FS-PODB cannot be claimed to be no-regret.

4.  Condition 2 holds for small $T$. Therefore, Theorem 5, 7 and 8 also hold for small $T$. I think the authors are misusing the $\mathcal{O}(\cdot)$ notations in these theorems. Note that a $\mathcal{O}(\sqrt{T})$ regret should mean that the algorithm incurs at most $k\sqrt{T}$ regret for all $T\geq T_0$ where $k, T_0$ are some constants. This is clearly not true for Theorem 5, 7 and 8.

5. In the stochastic setting, the regret obtained by the authors for large $T$ is $\mathcal{O}(T^{3/4})$ (Theorem 6). This should be the only result reported by the authors in the stochastic setting, and Theorem 5 should be discarded. The introduction and abstract should be changed accordingly. Surely, $\mathcal{O}(\sqrt{T})$ should not be highlighted as the primary result since it might mislead the readers. In the above light, the result of the authors seems weaker than Stradi et. al. (2024b). The claims accordingly have to be weakened in the introduction.

6. Similarly, Theorem 7, 8 should be discarded and only Theorem 9 should be kept in the adversarial setting with modified baseline. Since Theorem 7 was one of the main contributions of this paper, I am not sure what contribution of the paper remains.

7. The authors did not discuss the computational complexity of their algorithm in each iteration, although it was once of the main motivation mentioned in the introduction. In order to compute this complexity, the authors should take $S^H$ as the size of the state space since a general state space may not have a layered structure. The authors should compare their computational complexity with that of the earlier papers which they claimed to have more difficult problems to solve in each iteration.

**Questions:**

1. (Line 144): It seems like the occupancy measure, $q^{P, \pi}$ should be a function of $h$. However, it certainly is not. Am I missing something? Should there be an averaging over $h$ instead?

2. I see that the regret is computed using the optimal policy in hindsight. However, I believe a more sensible approach should have been to compute the optimal policy for each episode separately and then use that to compute regret. In summary, instead of using $T\mathrm{Opt}\_{\bar{r}, \bar{G}}$ as the baseline in regret computation, the term $\sum_{t} \mathrm{Opt}\_{r_t, G_t}$ should have been used. Any comment on why the second approach is not used?

3. Theorem 7: Can the roles of regret and constraint violation be reversed? In other words, is it possible to derive a $\mathcal{O}(\sqrt{T})$ regret bound while the constraints are satisfied up to some fractions?

4. Section 4.2.1: Shouldn't $\mathrm{Opt}^{\mathcal{W}}$ be a function of $t$ (since $G_t$ is used in the optimization instead of its time averaged value)?

---

### Author Response · Authors · 2024-11-15
**Part 1**

Dear Reviewers,

we would like to thank you for the effort in evaluating our work. We decided to answer you with an official comment since many of the concerns are shared by most of the Reviewers.

**On Condition 2**. We apologize for the typo, but the condition required by our algorithm to properly work is **$\rho\geq T^{-1/8}H\sqrt{112m}$**. We believe that this correction answers many of the Reviewer concerns about the correctness of our results. We already updated the corrected version of the paper.

**On the primal regret minimizer**

**1) Regret bound**

It is not correct to state that our primal algorithm cannot be defined as no-regret since $\Xi_t$ can be an increasing function in $T$. Indeed, the primal regret bound depends on the maximum payoff range in the interval (namely, $\Xi_{t_1,t_2}$). Nonetheless, it is **standard for online learning algorithms** to scale at least linearly in the payoff range, and it is standard in the no-regret definition to consider the payoff range a constant independent of $T$.


**2) Why policy optimization?**

The advantage of employing policy optimization approach in the primal regret minimizer is twofold.

First of all, this kind of procedures are computationally much more efficient. Indeed, policy optimization updates are solved by employing dynamic programming techniques, namely, the MDP is simply traversed to compute the state-action value function. On the contrary, occupancy-based methods require to solve convex programs at each episode, in order to project over the occupancy-measure decision space. Notice that, while convex programs can be approximated arbitrarily well in polynomial time, they require specific methods to be solved.

Furthermore, policy-optimization approaches are more suitable for guaranteeing the no-interval regret property. Indeed, we managed to show the no-interval regret property by introducing a fixed-share update in the policy optimization primal algorithm. This kind of update cannot be performed on the occupancy measure space, since, due to the transitions estimations, the center of the estimated occupancy measure space could lie outside the decision space induced by the true transition function. That is why, **(Stradi et al. 2024) primal algorithm cannot be employed to deal with bandit feedback, by just changing the estimator**.

**On the dependence of $\rho$ in the theoretical bound**

We believe that the dependence on $\rho$ **is not** a drawback in [Stradi et al. 2024]. Indeed, assuming Slater's condition to hold and attaining theoretical bounds the scales as the inverse of the Slater's parameter is standard in the literature of primal-dual methods.
In the following, some references.

["Exploration-Exploitation in Constrained MDPs", 2020]

["Learning Policies with Zero or Bounded Constraint Violation for Constrained MDPs", 2023]

["Provably Efficient Model-Free Algorithms for Non-stationary CMDPs", 2023]

["Truly No-Regret Learning in Constrained MDPs", 2024]

In general, it is also standard in the literature to assume the knowledge of the value of $\rho$ (see many of the references mentioned above), which is not the case in our work.

Thus, we focused on relaxing what we believe are the main disadvantages of [Stradi et al. 2024], namely, the full-feedback assumption and the per-round projection in the occupancy measure space.

---

### Author Response · Authors · 2024-11-15
**Part 2**

**On the contributions**

While our primal-dual scheme follows the one of [Stradi et al. 2024], we believe that the contribution of our work is clear.

First of all, the analysis of the primal regret minimizer is completely independent from [Stradi et al. 2024]. As previosuly specified, occupancy-measure based methods are not suitable for guaranteeing the no-interval regret property. Moreover, even if our primal regret minimizer shares similarity between [Luo et al., 2021], the algorithms are not equivalent. Our analysis and the one in [Luo et al., 2021] are substantially different in two fundamental aspects.
First, we show that our algorithm works with a dynamic payoff range, paying a linear dependence only in the maximum range. While this dependence is standard in online algorithms, this happens since the payoff range is known a priori. This is not the case in our setting. To avoid a quadratic dependence on the payoff range, we had to suitability modify the learning rate. Moreover, as highlighted in our work, our algorithm guarantees the no-interval regret property. Proving this result makes our analysis independent from the one of [Luo et al., 2021].
We underline that, showing the no-interval regret property in new settings (in our case, adversarial MDPs with bandit feddback) is of independent interest.

As a second aspect, notice that all the primal-dual analysis in [Stradi et al. 2024] is designed to work under full-feedback. Indeed, the Lagrangian function given to both the primal and the dual in our work, capture less information that the one in [Stradi et al. 2024], namely, the Lagrangian is built for the path traversed only.

Finally, notice that, differently from any prior work on CMDPs, we complement our analysis studying the regret w.r.t. an optimal solution which satisfies the constraints at each episode. While this baseline is weaker than the first one we propose, it has been largely studied in different settings w.r.t. ours, since it is the only one which allows to be no-regret when the constraints are adversarial. Specifically, this baseline has been studied in online convex optimization with constraints, thus, in single state environments with full-feedback (see, e.g., [``Tight Bounds for Online Convex Optimization with Adversarial Constraints", 2024]). Our results match theirs, showing that $\sqrt{T}$ regret and violations are attainable when only bandit feedback is available and for an MDP structure of the environment. This analysis is novel and independent from prior works.

We ask the Reviewer to let us know if further discussion on any concern is needed and we hope that we have properly addressed the Reviewer's concerns.

Thanks,

the Authors

---

### Meta-Review · Area_Chair_JFWA · 2024-12-07

**Metareview:**

This paper considers the online constrained Markov decision problem (CMDP) with bandit feedback under both stochastic and adversarial constraints. The CMDP considered is under the finite horizon episodic setting. The authors propose a best-of-both-worlds algorithm that can handle the considered problem setting with both stochastic and adversarial constraints. Based on the value of a constant $\rho$ that captures the "degree" that Slater's condition is satisfied, $O(T^{1/2})$ or $O(T^{3/4})$ regret/violation can be achieved.

After the author-reviewer discussion, there are still several issue that has yet been cleared.

First, even as a theoretical paper, this work lacks basic numerical evaluation of the result. Second, and most importantly, even after correcting the typo in Cond. 2, it still remains not well-justified. According to the defination, $\rho$ is in fact not a constant. In stochastic setting it relies on $\bar{G}:=(G_1+\cdots+G_T)/T$. In adversarial setting it relies on all $G_t, t\leq T$ and involves a minimum over $t$. That is $\rho$ should in fact be $\rho_T$ and is changing in each step. The current Cond. 2 is misleading in that $\rho$ looks like a constant and Cond.2 "seems to hold when T is large enough". However, as $\rho_T$ is changing in each step (even monotonically decreasing for adversarial setting), there lacks justification on why Cond. 2 should eventually hold.

Overall, the AC would give a rejection to the paper.

**Additional Comments On Reviewer Discussion:**

The main issue raised by the reviewers are about the validity of Condition 2, where the issue terms out to be a typo. However, even after fixing the typo, it still remains problematic as it is presented like making assumptions about a constant, which is in fact a changing number depending on all iterations and random feedback from the environments.

---

### Decision · Program_Chairs · 2025-01-22

Reject